# Boosting photocatalytic hydrogen production from water by photothermally induced biphase systems

Shaohui Guo [1], Xuanhua Li[1✉], Ju Li[2] & Bingqing Wei [3✉]

Solar-driven hydrogen production from water using particulate photocatalysts is considered the most economical and effective approach to produce hydrogen fuel with little environmental concern. However, the efficiency of hydrogen production from water in particulate photocatalysis systems is still low. Here, we propose an efficient biphase photocatalytic system composed of integrated photothermal–photocatalytic materials that use charred wood substrates to convert liquid water to water steam, simultaneously splitting hydrogen under light illumination without additional energy. The photothermal–photocatalytic system exhibits biphase interfaces of photothermally-generated steam/photocatalyst/hydrogen, which significantly reduce the interface barrier and drastically lower the transport resistance of the hydrogen gas by nearly two orders of magnitude. In this work, an impressive hydrogen production rate up to 220.74 $\mu$mol h$^{-1}$ cm$^{-2}$ in the particulate photocatalytic systems has been achieved based on the wood/CoO system, demonstrating that the photothermal–photocatalytic biphase system is cost-effective and greatly advantageous for practical applications.

[1] State Key Laboratory of Solidification Processing, Center for Nano Energy Materials, School of Materials Science and Engineering, Northwestern Polytechnical University, Xi'an, China. [2] Department of Nuclear Science and Engineering, Massachusetts Institute of Technology, Cambridge, MA, USA. [3] Department of Mechanical Engineering, University of Delaware, Newark, DE, USA. ✉email: lixh32@nwpu.edu.cn; weib@udel.edu

Solar-driven hydrogen production from water is a potentially efficient way to address the environmental problems and global energy crisis of fuel production. In particular, hydrogen gas has a high energy capacity (143 MJ kg$^{-1}$) and releases no toxic emissions[1]. Therefore, an efficient and rapid photocatalytic hydrogen-production method is urgently needed[2,3]. There are three main types of solar-driven hydrogen production systems: particulate photocatalysis, photovoltaic-assisted electrolysis (PV-E), and photoelectrochemical cells (PEC)[2], where the particulate photocatalysis is predicted to be more cost-effective than the other two systems[4]. Unfortunately, the solar to hydrogen conversion efficiency in particulate photocatalysis remains low though many strategies, including structural and defect engineering, plasmonic effects, and elemental doping, have been discussed to improve photocatalysts' optical absorption and photo-induced charge separation and transport[5–7].

In this work, from the phase-interface perspective, we design an efficient and cost-effective photocatalytic system composed of integrated photothermal–photocatalytic materials that can easily convert liquid water to water steam via photothermal transpiration effect with charred wood substrates. And the steam is simultaneously split into hydrogen by the photocatalysts loaded on the wood under light illumination without additional energy. The design exhibits biphase interfaces of self-generated steam/photocatalyst loaded on the charred wood substrates/hydrogen gas. Our strategy of the photothermally induced biphase interfacial feature differs from previous studies of the room-temperature vapor in moisture environment to reduce the catalysts corrosion (the humidity was realized through a complex microfluidic microreactor[8–10], convection effect[11], and hydrophobic effect[12]) and plasmonic thermal effects[13] and near-infrared photothermal effects[14,15] in the triphase interfaces of liquid water/photocatalyst/hydrogen. This photothermal–photocatalytic biphase system kinetically lowers the hydrogen gas's transport resistance by nearly two orders of magnitude to allow the easy escape of hydrogen gas from the system. It also thermodynamically reduces the interface barrier in the adsorption process of gas-phase water molecules to photocatalysts. In this work, such a biphase system significantly improves the photocatalytic hydrogen production rate up to 220.74 μmol h$^{-1}$ cm$^{-2}$ for the wood/CoO system and 3271.49 μmol h$^{-1}$ cm$^{-2}$ for the wood/CuS–MoS$_2$ heterophotocatalyst.

## Results

### Constructing a photothermal–photocatalytic system on charred wood.
A photothermal–photocatalytic system was skillfully designed and implemented by applying natural wood to generate water steam via photothermal transpiration under the light illumination simulated by a solar simulator at AM 1.5 G illumination (100 mW cm$^{-2}$)[16–18]. Also serving as the substrate for the photocatalytic reaction, a wood slice was cut from a tree perpendicular to its growth direction, and the surface of the wood slice was carbonized by a simple heating process for improving the steam generation with a high solar-to-steam-conversion efficiency of 46.90% (Fig. 1a and Supplementary Figs. 1–4)[17]. CoO nanoparticles (NPs), as a typical photocatalyst, were spin-coated on the carbonized wood slices to construct the wood/photocatalyst photothermal–photocatalytic system (here, the wood/CoO system), as shown in Fig. 1a. The monodispersed CoO NPs are ~50 ± 5 nm in diameter, and the CoO lattice fringes (with a d-spacing of 0.24 nm) are assigned to the (111) lattice planes of CoO, as shown in Supplementary Fig. 5[19]. The light absorption peak of the CoO NPs locates at 550 nm (Supplementary Fig. 6).

The CoO NPs are distributed ~2 mm along the walls of the wood microchannels, as evidenced by the Raman spectra at different depths from the top surface (Fig. 1b), where only four

Raman spectra taken with an interval of 500 μm from the surface show the CoO Raman characteristic peaks at 473.6 and 540.9 cm$^{-1}$ [20]. When the wood/CoO system floats in the water, the immersion depth of the wood in the water is about 2 mm (Fig. 1c), indicating that the photocatalysts are not directly soaked in the liquid-phase water.

After CoO NPs coating on the wood, the wood/CoO system shows high light absorbance from 300 to 1000 nm compared to that of the pure wood, as shown in Fig. 1d, implying that the wood/CoO system can effectively utilize solar energy. Under light illumination, the surface temperature of the wood/CoO system is about 325 K (Fig. 1e), and the adhered photocatalysts become covered with steam produced by the photothermal transpiration in the wood interior. Simultaneously, the photo-induced electrons participate in the hydrogen evolution reaction at the photocatalytic active sites, and photo-induced holes participate in the H$_2$O$_2$ generation (Fig. 1a). It should be noted that the local temperature of the CoO NP is estimated to be 346 K based on the potential (Fig. 1f, g, and Supplementary Fig. 7)[21], which is higher than the global temperature (325 K, in Fig. 1e) because of the nanoscale effect. It is speculated that a higher local temperature is beneficial to enhance the photocatalytic reaction efficiency.

We investigated the effect of CoO NPs mass loading on the photocatalytic hydrogen gas production rate in the wood/CoO system (Fig. 2a). An optimized mass loading of about 38 mg cm$^{-2}$ CoO NPs has been identified based on the experimental results. The photocatalytic H$_2$ evolution rate in the wood/CoO system with 38 mg cm$^{-2}$ CoO NPs loading is about 5776 μmol h$^{-1}$ g$^{-1}$ (i.e., 220.74 μmol h$^{-1}$ cm$^{-2}$), 17 times higher than that of the triphase CoO NPs (337 μmol h$^{-1}$ g$^{-1}$, agrees well with the values reported under similar conditions)[19], as shown in Fig. 2b. For clarification, the photocatalytic activity of wood alone was measured under the same condition (Supplementary Fig. 8), and no trace of hydrogen gas and oxygen gas were detected after 2 h of reaction, indicating that the wood does not have photocatalytic activity. Moreover, we also studied the effect of solar intensity on the photocatalytic response of the wood/CoO system. As shown in Fig. 2c, the rate of hydrogen evolution grows with the increase of solar intensity but not a linear relation. This is mainly because of the temperature rising on the wood/CoO surface caused by the increase in solar intensity (Supplementary Fig. 9). A higher temperature can exponentially improve the rate of hydrogen evolution, as evidenced in the following section.

In addition, the biphase wood/CoO system exhibits superior stability in photocatalytic activity. The long-period photocatalytic hydrogen production measurement with the wood/CoO system was conducted for 5 days (Fig. 2d). On day 1, the initial hydrogen production rate in 1 h is 221.56 μmol h$^{-1}$ cm$^{-2}$, and the average hydrogen production rate during 8 h reaction is 194.14 μmol h$^{-1}$ cm$^{-2}$. On day 5, the average hydrogen production rate during 8 h reaction is 174.73 μmol h$^{-1}$ cm$^{-2}$. Thus, after 5 days (40 h) test, the photocatalytic hydrogen evolution performance maintains about 90%, exhibiting that photocatalytic stability can be significantly improved through the wood/catalysts system compared to that in the previous work, which only holds 1 h of reaction[22]. We also studied the morphological stability of the wood/CoO system. After the photocatalytic reaction, the CoO NPs remain well attached to the wood matrix structure, further confirming the stability of the wood/CoO system (Fig. 2e, f, Supplementary Figs. 10 and 11). There is little difference in the reflection spectra and X-ray photoelectron spectroscopy (XPS) spectra of the wood/CoO system before and after the photocatalysis process (Supplementary Figs. 12 and 13). Besides, the inductively coupled plasma emission (ICP) and ultraviolet–visible (UV–Vis) spectra of the bulk water in the wood/CoO system after the photocatalytic reaction have been measured (Supplementary

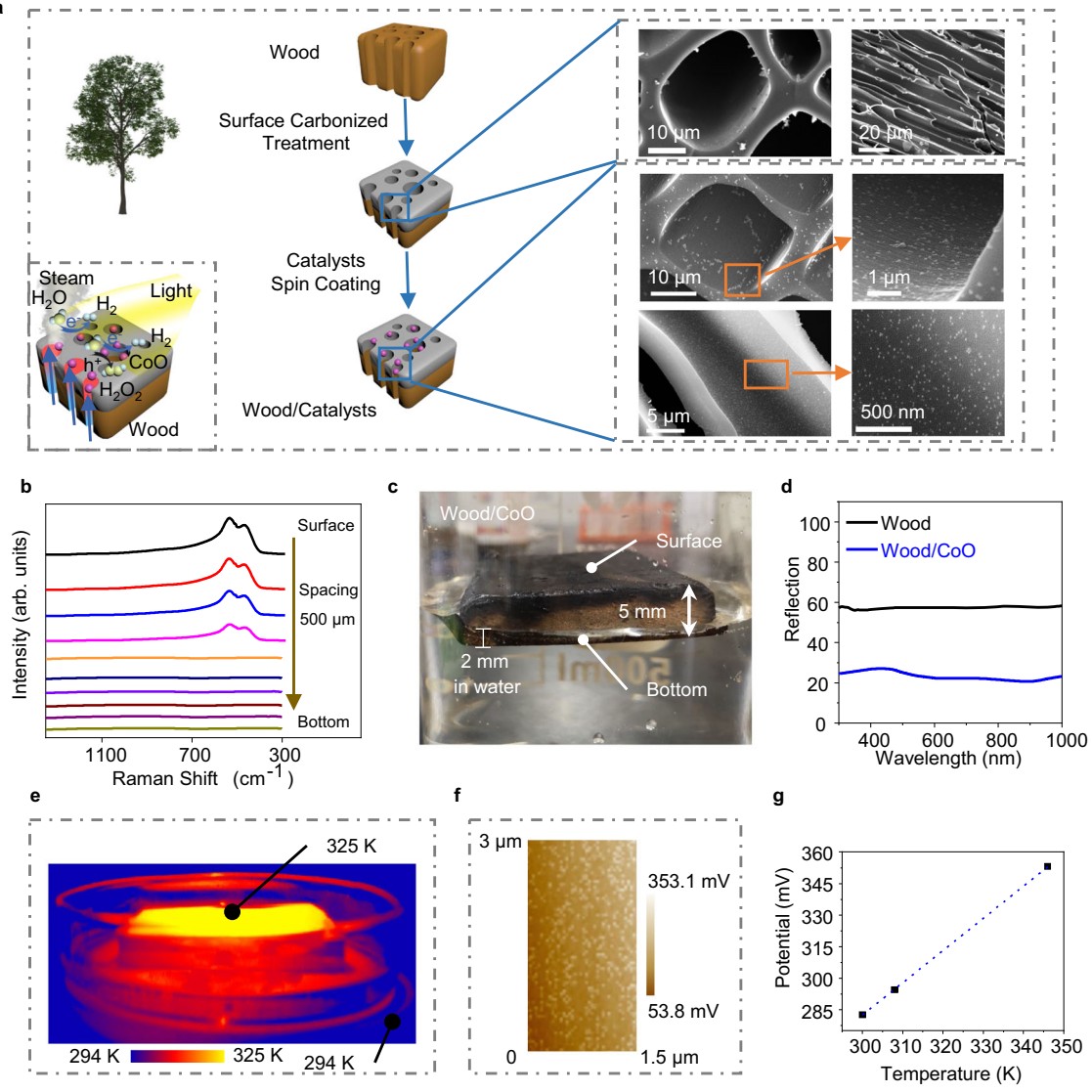

**Fig. 1 The designed wood/photocatalyst biphase photothermal–photocatalytic system. a** Schematic of the fabrication process of the wood/photocatalyst structure that generates the water steam and catalyzes its splitting for hydrogen evolution. **b** Raman spectra taken at different depths along the cross-section of wood/CoO microchannels with an interval of 500 μm. **c** The photo of the wood/CoO system floating on water. **d** Reflection spectra of the wood and wood/CoO systems. **e** Infrared radiation thermal image from the wood/CoO system under light illumination. **f** The potential of wood/CoO under 100 mW cm$^{-2}$ light illumination. **g** The estimated local temperature of CoO NPs through the measured potential.

Fig. 14 and Supplementary Table 1). There are few amounts of element Co in the bulk water based on the ICP measurement results and the absorption spectrum, exhibiting the wood/CoO system's excellent stability. It can be concluded that the photothermal–photocatalytic system displays a significant advantage in substantially enhancing the H$_2$ evolution rate from water splitting. It is noticed that the ratio of photocatalytic H$_2$ and O$_2$ production is not equal to 2:1 (Supplementary Figs. 15 and 16), mainly due to the generation of H$_2$O$_2$ by-product in the photocatalytic process (Supplementary Fig. 17).

**Understanding the phase-interface effect on catalytic performance**. From the phase-interface perspective, the photothermal–photocatalytic system exhibits biphase interfaces of photothermally-generated steam/photocatalyst/hydrogen gas. To understand the phase-interface effect on the photocatalytic performance, we conducted experiments with a biphasic photocatalytic system containing injected water steam/solid photocatalysts (Fig. 3a and c). Water steam was injected and controlled by a steam flowmeter into

a transparent reactor, where CoO NPs powder catalysts were placed on the surface of a filter paper, and no sacrificial agent was added to the photocatalytic system. Under light illumination, the steam in the reactor was photocatalytically converted to H$_2$, which was detected by the gas chromatography (GC) (Fig. 3c). For comparison, the liquid/solid/gas triphase system of water/photocatalyst/hydrogen in common photocatalytic hydrogen evolution reaction has also been included. As shown in Fig. 3b, hydrogen bubbles are generated when the solid photocatalysts are interacting with liquid water under light illumination. The produced hydrogen gas is then collected by passive transport against the liquid water phase.

Hydrogen production in the biphase photocatalytic reaction system was evaluated with different flow rates of water steam (from 5 to 88 ml h$^{-1}$) injected into the reactor chamber (Fig. 3c, d, Supplementary Figs. 18 and 19). The rate of hydrogen production from steam increases along with the increase of steam flow rate from 5 to 62 ml h$^{-1}$. When the steam flow rate further grows, the hydrogen production rate is stabilized because the quantity of

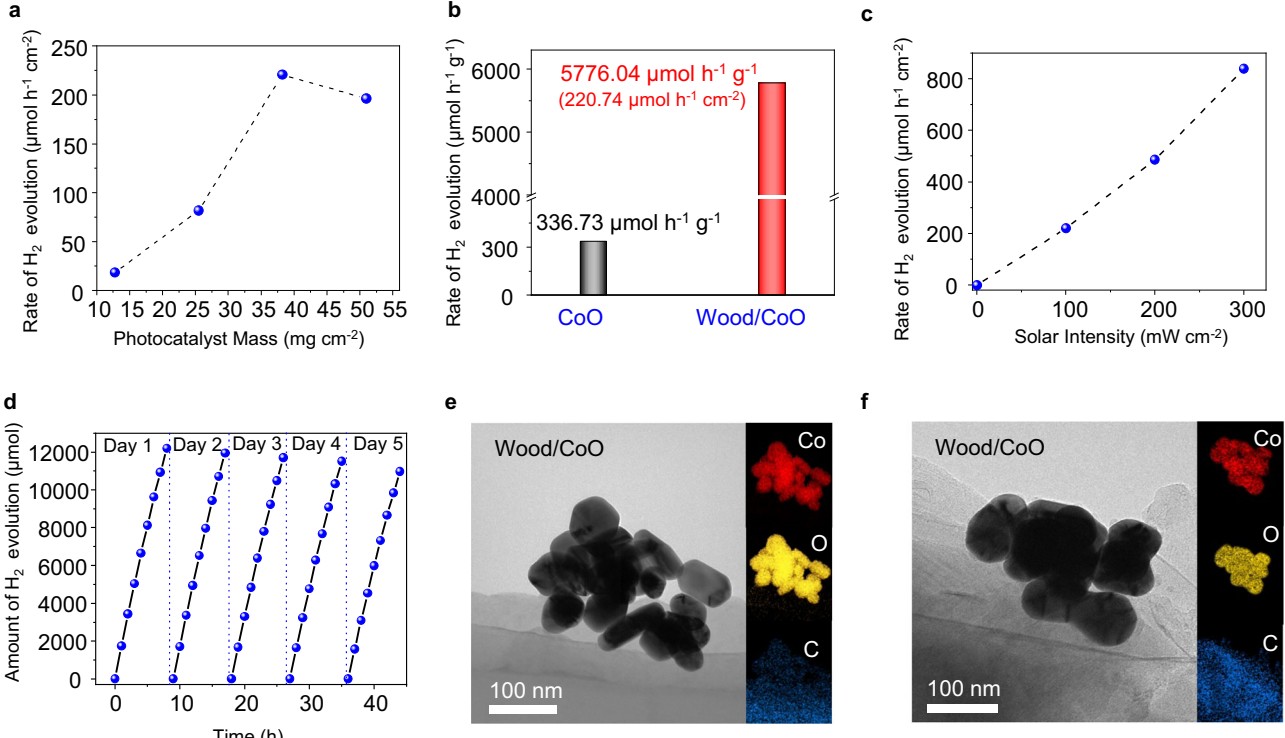

**Fig. 2 Photocatalytic hydrogen evolution in the designed wood/photocatalyst biphase photothermal–photocatalytic system. a** Mass loading-dependent photocatalytic hydrogen gas production rates for the wood/CoO system (area: 7.85 cm$^2$). **b** Rate comparison of H$_2$ evolution in the triphase CoO NPs system and wood/CoO biphase photothermal–photocatalytic system (area: 7.85 cm$^2$, mass: 0.3 g). **c** Rate of H$_2$ evolution versus solar intensity from the wood/CoO system (area: 7.85 cm$^2$, mass: 0.3 g). **d** Time-dependent photocatalytic hydrogen gas production of the wood/CoO system for 5 days. The light source is a solar simulator at AM 1.5 G illumination (100 mW cm$^{-2}$). **e** TEM images and EDS element mapping of CoO NPs attached to the walls of the wood microchannels before the photocatalytic reaction. **f** TEM images and EDS element mapping of CoO NPs attached to the walls of the wood microchannels after the photocatalytic reaction.

water molecules reaches saturation during the photocatalytic reaction. At the optimal flow rate (i.e., 62 ml h$^{-1}$), the maximum hydrogen evolution rate is ~6200 μmol h$^{-1}$ g$^{-1}$, 18 times higher than that in the triphase reaction system (337 μmol h$^{-1}$ g$^{-1}$). The biphase photocatalytic system also shows excellent stability of the photocatalytic reaction, as shown in Fig. 3e. After three cyclic measurements, the amount of H$_2$ evolution concurs with that in the first measurement. And the morphology and absorption spectra of CoO NPs after the photocatalytic reaction also keep unchanged, confirming the excellent stability of the photocatalyst (Supplementary Figs. 5, 20, and Fig. 3f).

The main factors governing the photocatalytic hydrogen evolution in the biphase reaction system are the temperature and the state of water in comparison to the triphase reaction system. Figure 4a shows the temperature-dependent of the photocatalytic hydrogen evolution rate with the CoO NPs photocatalyst in the triphase reaction system. As the reaction temperature increases from 298 K to close to 373 K, the hydrogen evolution rate monotonically increases from 336.73 to 1968.9 μmol h$^{-1}$ g$^{-1}$ (note that 373 K is the steam-conversion temperature of liquid water). It should be noted, however, no trace hydrogen is detected after 2 h of reaction at near 373 K if light illumination is not applied, implying that the catalytic reaction cannot be thermally triggered (Supplementary Fig. 21). Furthermore, the relationship between the rate of H$_2$ evolution reaction $V$ and the reaction temperature $T$ can be well-fitted with the Arrhenius equation:

$$V = 3748519.38 e^{-\left(\frac{23023}{8.314 * T}\right)} \quad (1)$$

According to Eq. (1), the activation energy for the hydrogen production over CoO was deduced to be 23.023 kJ mol$^{-1}$. The activation energy is a key indicator to reflect whether photocatalytic hydrogen evolution reaction occurs easily. The smaller the activation energy is, the easier the hydrogen production process will become. Therefore, a low activation energy here indicates that the hydrogen production process is easily conducted on the CoO NPs. Furthermore, the H$_2$ evolution rate at 373 K is estimated to be 2236.76 μmol h$^{-1}$ g$^{-1}$. However, it is much lower than the H$_2$ evolution rate (6200.42 μmol h$^{-1}$ g$^{-1}$, see Fig. 3d) in the biphase reaction system at the same temperature of 373 K, indicating that the temperature effect on improving the H$_2$ evolution rate is limited although a higher reaction temperature does promote the photocatalytic hydrogen-evolution reaction. Thus, in addition to the reaction temperature, the state of water plays a crucial role in enhancing the hydrogen evolution of the biphasic reaction system.

The temperature effect can be systematically analyzed from two aspects: thermodynamics and kinetics. Three reaction steps, including the adsorption of water molecules, the adsorption of hydrogen atoms, and the hydrogen gas production in photocatalytic reaction, have been involved. First, the Gibbs energy in the triphase system has been calculated at 298 and 373 K, where the pure CoO structure without any group is used to simulate the CoO status in the neutral environment because the pH value of the reactant water is approximately equal to 7. As shown in Fig. 4b, the change of reaction temperature from 298 K to 373 K influences the first and second steps. The Gibbs energy of the water molecule adsorption process at 298 K is about 0.426 eV, and it is about 0.145 eV for the hydrogen adsorption process.

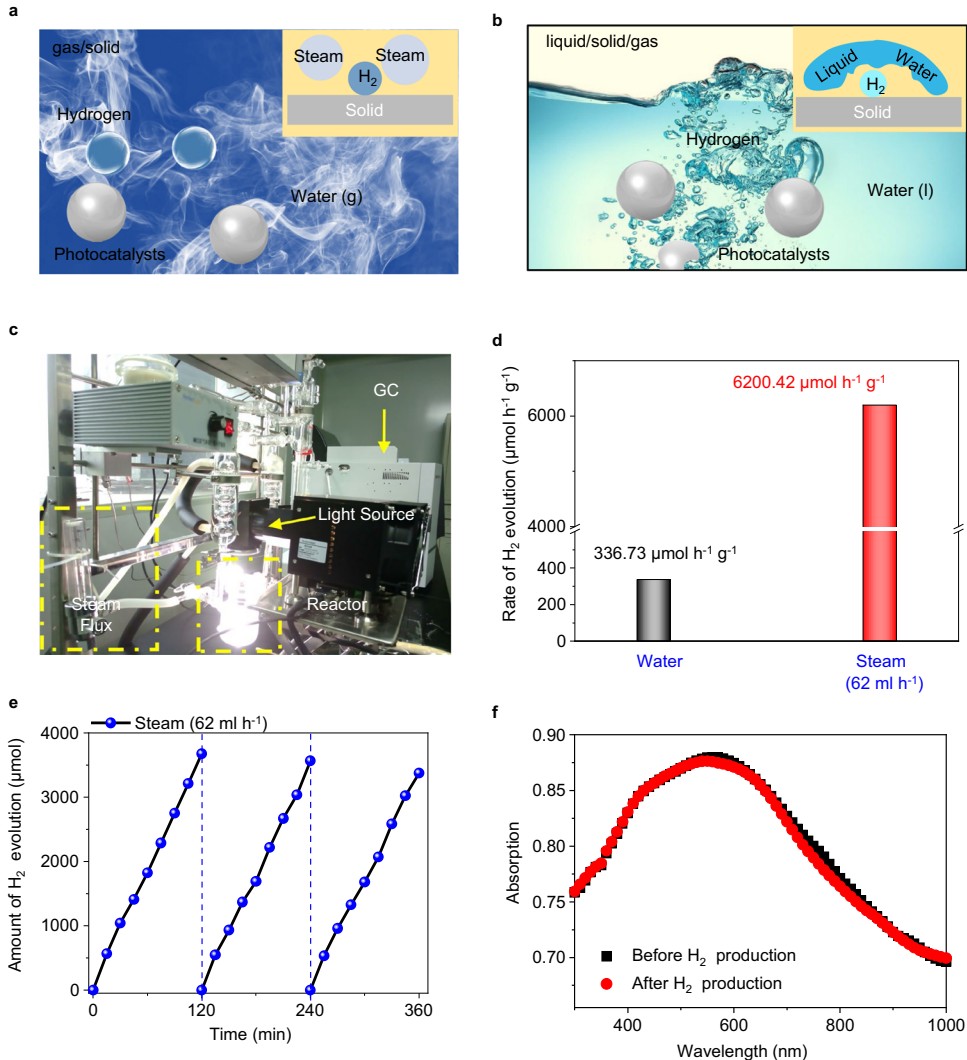

**Fig. 3 Schematic and photocatalytic hydrogen evolution performance in the biphase and triphase reaction systems. a** Schematic of the biphase system with water steam/solid photocatalysts/hydrogen gas. **b** Schematic of a typical triphase system consisting of liquid water/solid photocatalysts/hydrogen gas. **c** Photo of the biphase photocatalytic system. **d** Photocatalytic hydrogen gas production rate from the triphase and biphase reaction systems. **e** Time-dependent photocatalytic hydrogen gas production from water steam with a flow rate of 62 ml h$^{-1}$. The light source is a solar simulator at AM 1.5 G illumination (100 mW cm$^{-2}$). **f** The absorption spectra of CoO NPs before and after hydrogen gas production.

Comparatively, they are reduced to 0.331 and −0.054 eV, respectively, at 373 K. As expected, the high reaction temperature in the triphase photocatalytic system would thermodynamically favor the water molecule adsorption process.

In addition to kinetically promote the transport of water molecules, high temperatures will reduce hydrogen transport resistance as well so that the photocatalytic reaction rate can be accelerated. This can be evidenced by the hydrogen gas diffusion coefficient $D_L$ in a liquid-phase environment, calculated by the Stokes–Einstein equation:

$$D_L = 7.4 \times 10^{-8} \frac{T(\psi_{H_2O} M_{H_2O})^{0.5}}{\mu V_{H_2}^{0.6}} \qquad (2)$$

where $T$ is the temperature, $\psi_{H_2O}$ (=2.26) is the "association" parameter of the solvent water, $M_{H_2O}$ and $\mu$ denote the molecular weight and viscosity of water, respectively, and $V_{H_2}$ is the molar volume of hydrogen. When the temperature is increased from 298 to 373 K, the hydrogen gas diffusion coefficient $D_L$ is increased. Thus, hydrogen transport resistance is slightly decreased.

A more significant effect on promoting the photocatalytic hydrogen-evolution reaction comes from the state change of the water phase. When the water phase changes from liquid to steam at the same temperature (373 K), interestingly, the first and second step of the photocatalytic reaction (i.e., the water molecule adsorption process and the hydrogen adsorption process) has been significantly influenced. The Gibbs energy of the water molecule adsorption process substantially decreases from 0.331 eV in the triphase system to −0.212 eV in the biphase system, and it also reduces (−0.054 vs. −0.007 eV) for the hydrogen adsorption process (Fig. 4c), indicating that the water molecule adsorption process and hydrogen adsorption process in the biphase system become much more comfortable than that in the triphase system. Kinetically, the hydrogen gas diffusion coefficient $D_G$ in the gaseous environment can be calculated by the Chapman–Enskog theory:

$$D_G = \frac{A * T^{3/2} \sqrt{\frac{1}{M_{H_2}} + \frac{1}{M_{H_2O}}}}{P * \sigma^2 * \Omega} \qquad (3)$$

where $A$ (=1.858 × 10$^{-3}$) is an empirical coefficient[23,24], $M$ is the

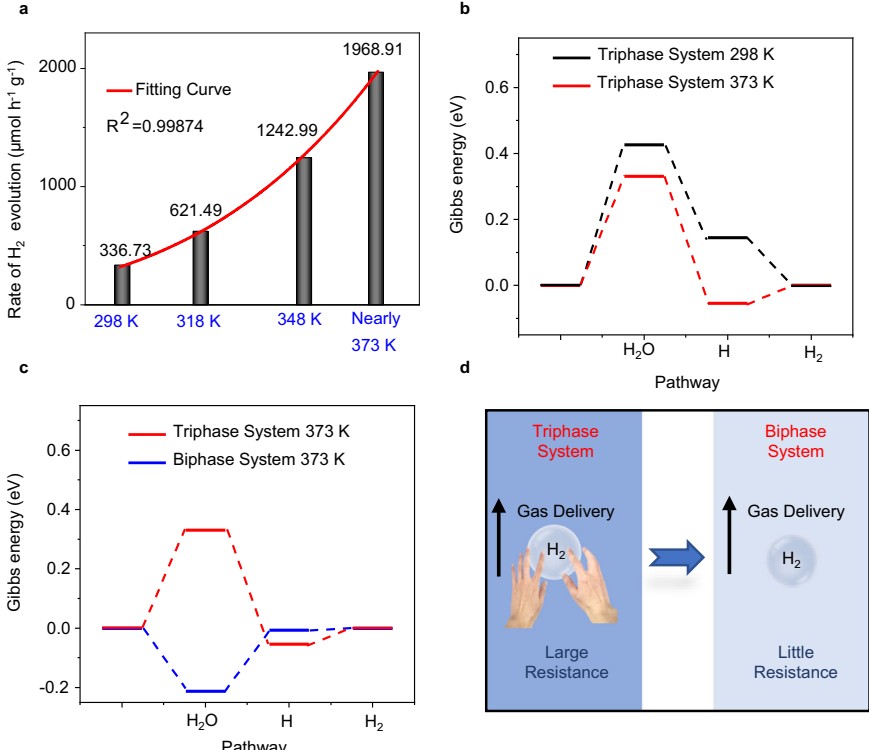

**Fig. 4 The factors governing the biphase photocatalytic hydrogen evolution: temperature and liquid-to-gas-phase changes of water. a** The photocatalytic hydrogen evolution rate with the CoO NPs versus the reaction temperature in the triphase reaction system. (Here, the background pressure was maintained at 5 kPa). **b** Gibbs energy of a photocatalytic reaction in the triphase system with different temperatures over the pure CoO surface. The photocatalyst is CoO NPs. **c** Gibbs energy of a photocatalytic reaction in the triphase system (373 K) in comparison with the biphase system (373 K) over the pure CoO surface. The photocatalyst is CoO NPs. **d** Schematic of the hydrogen transport resistances in the liquid- and gas-phase environments.

molar mass, $P$ is the pressure in the system, $\sigma$ is the average collision diameter, and $\Omega$ is a temperature-dependent collision integral. The produced hydrogen bubbles experience frictional resistance in adjacent interlayers from relative motion with the environmental particles[25]. Owing to the interfacial frictional resistance, the hydrogen-gas diffusion coefficients differ significantly in the liquid and steam water phases. In the liquid water phase, $D_L$ is $(4.99-5.06) \times 10^{-5}$ at reaction temperatures near 373 K (i.e., 368–373 K), based on the Stokes–Einstein equation (Eq. (2)), whereas, in the steam phase, $D_G$ is $2.65 \times 10^{-3}$ at 373 K, two orders of magnitude higher than that in the liquid water. Therefore, when the produced hydrogen gas passes through the liquid water before being liberated, it is greatly resisted by the environmental liquid water molecules. By contrast, the hydrogen bubbles in the biphase system pass through the gas water molecules with much less resistance. Figure 4d schematizes the hydrogen transport resistances in the liquid- and gas-phase environments.

**The universality of the photothermal–photocatalytic system.** In addition to the exemplary wood/CoO system, the photothermal–photocatalytic system can also extend to other photocatalysts. To demonstrate the universal feature of the photothermal–photocatalytic biphase system, different photocatalysts, i.e., $MoS_2$, $C_3N_4$, and $TiO_2$ were, respectively, spin-coated on the carbonized wood slices to construct wood/photocatalyst architectures (Fig. 5a–c and Supplementary Figs. 22–24). The $MoS_2$, $C_3N_4$, and $TiO_2$ photocatalysts are all uniformly distributed and attached to the microchannel walls of the wood. The particulate photocatalytic hydrogen-evolution reactions were carried out in all the wood/photocatalyst reaction systems (Fig. 5d and Supplementary Fig. 25).

All of the photocatalysts realize photothermal–photocatalytic hydrogen production, but no oxygen is detected at the same time because of the difficulty in downshifting the valence band positions (e.g., $MoS_2$, $C_3N_4$) and complex surface deformation reaction (e.g., $TiO_2$)[26–28]. The $H_2$ average production rates of the wood/$MoS_2$, wood/$C_3N_4$, and wood/$TiO_2$ photothermal–photocatalytic systems are 155.77, 95.54, and 59.87 $\mu mol\,h^{-1}\,cm^{-2}$, respectively. For each photocatalyst, the apparent quantum yield (AQY) of the photothermal–photocatalytic biphase system dominates compared with the previously reported photocatalyst systems (Fig. 5e)[29–41], and the measured data are listed in Supplementary Table 2.

In addition to the monothetic particulate photocatalysts, a heterojunction photocatalyst, i.e., $CuS–MoS_2$, has also been introduced to the photothermal–photocatalytic systems to verify the universality (Supplementary Fig. 26). Similar to the monothetic particulate photocatalysts, the $CuS–MoS_2$ photocatalyst has adhered to the microchannel walls of the wood matrix, as shown in Fig. 5f. The photocatalytic $H_2$ average production rate of the biphase wood/$CuS–MoS_2$ photothermal–photocatalytic system reaches up to 85,604 $\mu mol\,h^{-1}\,g^{-1}$ (Supplementary Fig. 27), 16 times that of the triphase $CuS–MoS_2$ photocatalyst (5350 $\mu mol\,h^{-1}\,g^{-1}$) (Supplementary Fig. 28). It is noted that no photocatalytic oxygen gas was produced because of the energy band positions of the $CuS–MoS_2$ photocatalyst (Supplementary Fig. 29)[42]. It is speculated that the photo-induced holes react with some S ions from CuS/$MoS_2$ catalyst as shown based on the XPS results (Supplementary Fig. 30)[43]. Figure 5g summarizes the $H_2$ evolution rates of typical particulate photocatalysts reported to date. The $H_2$ evolution rates were 70,000, 64,426, and 11,090 $\mu mol\,h^{-1}\,g^{-1}$ in InP/ZnS[44], PTB7-Th/EH-IDTBR NPs[45], and 2D/2D NiS/Vs-ZnIn$_2$S$_4$/WO$_3$[46], respectively. They were 23,410 and 16,300 $\mu mol\,h^{-1}\,g^{-1}$ based on the

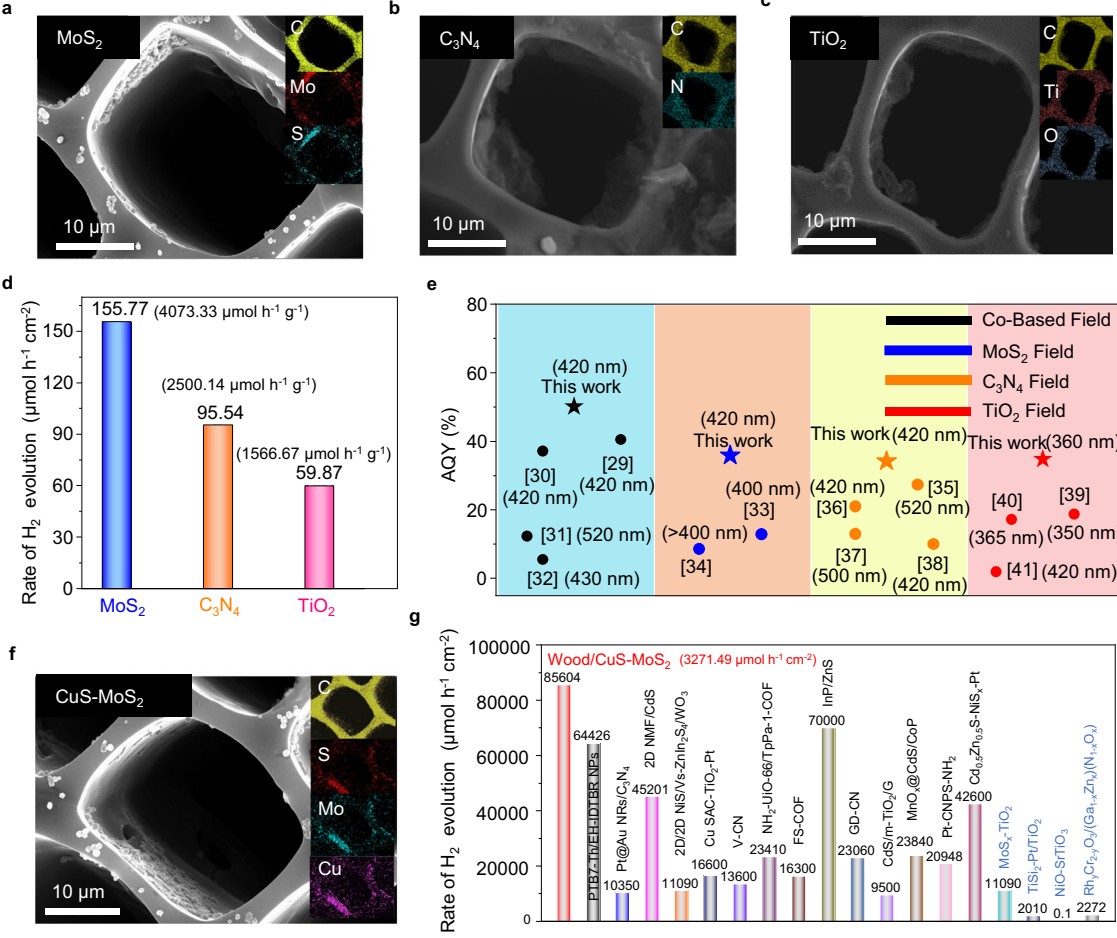

**Fig. 5 The universal feature of the photothermal–photocatalytic biphase system for photocatalytic hydrogen evolution from water. a** SEM images and EDS mapping of the wood/MoS₂. **b** SEM images and EDS mapping of the wood/C₃N₄. **c** SEM images and EDS mapping of the wood/TiO₂ systems. **d** H₂ evolution rates of the wood/MoS₂, wood/C₃N₄, and wood/TiO₂ systems. **e** Comparison of the AQY with literature in different particulate photocatalytic systems of TiO₂, C₃N₄, MoS₂, and Co-based photocatalysts, respectively. The numbers are the reference numbers and light wavelength. The maximum AQY data in literature and measurement results are presented for comparison. **f** SEM image and EDS mappings of the wood/CuS–MoS₂ photothermal–photocatalytic system. **g** Comparison of the H₂ evolution rate of different particulate photocatalytic systems reported to date. The blue fonts on the right represent the photocatalytic reaction in the gas-phase water environment. The light source is a solar simulator at AM 1.5 G illumination (100 mW cm⁻²).

COF materials[47,48]. The CdS-based photocatalysts including 2D NMF/CdS[49], $Cd_{0.5}Zn_{0.5}S-NiS_x-Pt$[7], $MnO_x@CdS/CoP$[50], and CdS/m-TiO₂/G[51] achieved a H₂ evolution rate of 45,201, 42,600, 23,840, and 9500 μmol h⁻¹ g⁻¹, respectively. An H₂ evolution rate of 16,600 μmol h⁻¹ g⁻¹ was reported on the Cu SAC-TiO₂-Pt photocatalyst[52]. The optimized C₃N₄-based heterojunction photocatalysts, including GD-CN[53], Pt-CNPS-NH₂[54], V-CN[55], and Pt@Au NRs/C₃N₄[56] achieved an H₂ evolution rate of 23,060, 20,948, 13,600, and 10,350 μmol h⁻¹ g⁻¹, respectively. In the reports involving vapor phase water, the photocatalytic hydrogen production rates were relatively low[12,57,58], and the leader was 11,090 μmol h⁻¹ g⁻¹ based on the MoSₓ-TiO₂ hybrid[11]. The photothermal–photocatalytic system, i.e., the wood/CuS–MoS₂ device, outperforms all of these photocatalysts with an H₂ evolution rate of 85,604 μmol h⁻¹ g⁻¹ (i.e., 3271.49 μmol h⁻¹ cm⁻²) without any external assistance, e.g., sacrificial agents, photovoltaic or photoelectrochemical assistance, demonstrating that the photothermal–photocatalytic biphase system can substantially enhance the photocatalytic hydrogen evolution from water.

Moreover, the photothermal–photocatalytic biphase system is promising for practical applications because it can easily be realized through the transpiration process of wood loaded with particulate photocatalysts. This natural process converts liquid water to steam under the same light illumination without additional energy input. As an example of a demonstration, the wood/CuS–MoS₂ was put in a reaction cell filled with simulated seawater (Supplementary Fig. 31), and the hydrogen collector was connected to the gas outlet. When exposed to natural sunlight, the hydrogen collector exhibits a visible bulge after 2 h of reaction. Although the salts (e.g., NaCl) in the seawater are possible to adhere to wood tunnels to clog the matrix structures, which lead to a decrease in steam production during evaporation[59], the H₂ production rate in this exemplary test is about 37,219 μmol h⁻¹ g⁻¹ (i.e., 1422.38 μmol h⁻¹ cm⁻²) (measured by GC), confirming the strong photocatalytic ability in a seawater environment. And after 6 h of reaction, the H₂ production rate remains consistent with that from the first test, exhibiting excellent photocatalytic stability.

## Discussion

We have designed and demonstrated an integrated photothermal–photocatalytic system that helps achieve the dominant photocatalytic hydrogen evolution rate of 85,604 μmol h⁻¹ g⁻¹ (i.e., 3271.49 μmol h⁻¹ cm⁻²) among the particulate photocatalysts.

Such excellent performance was achieved by replacing the traditional triphase photocatalytic interfaces (liquid water/photocatalyst solid/hydrogen gas) with the biphase photocatalytic interfaces (photothermally-generated water steam/photocatalysts loaded on charred wood substrates/hydrogen gas). The wood carrier functions simultaneously as the photocatalyst substrate as well as the steam generator under solar light, which is significantly advantageous for practical applications. This photothermal–photocatalytic system reduced the barrier of the water molecule adsorption process and minimized the delivery resistance of the produced hydrogen gas, enabling efficient and environmentally safe fuel for next-generation applications on an industrial scale.

## Methods

**Synthesis of CoO, MoS$_2$, C$_3$N$_4$, TiO$_2$, and CuS–MoS$_2$ photocatalysts**. CoO NPs were fabricated by a heating process with the hydrothermal method and a tube furnace. In all, 2 g of Co(CH$_3$COO)$_2$·4H$_2$O powder was added to a mixed solvent with 8 ml n-octanol and 32 ml ethanol by stirring for 3 h. After that, the mixture was transferred to a 50 ml Teflon-lined stainless steel autoclave, and then heated at 200 °C for 6 h. When the autoclave was cooled down to room temperature, the powders were placed in a quartz tube furnace. The tube was filled to ambient pressure with Ar gas flowing at 240 s.c.c.m. The Ar flow rate and 1 atm. pressure were maintained throughout the preparation process. The tube was continuously heated from 25 °C to 600 °C in 3 h. After maintaining the tube furnace at 600 °C for 5 h, the tube was cooled to room temperature over 5 h. The obtained powders were then dispersed in pure water, and the CoO NPs were obtained by centrifugation at 1677 × $g$ for 10 min (Anke TGL-15B Centrifuger). After that, the prepared photocatalysts (0.3 g) were added to the surface of the filter paper by spin coating at 500 rpm for 20 s. Then the filter paper with photocatalyst was taken into the oven at 40 °C for 1 h.

For the MoS$_2$ synthesis, solutions of 2.0 mmol Na$_2$MoO$_4$ and 4.0 mmol L-cysteine were sterilized in a 50 ml Teflon-lined stainless steel autoclave. The autoclave was heated at 200 °C for 12 h and naturally cooled to room temperature, obtaining the MoS$_2$ solution. For the C$_3$N$_4$ synthesis, 0.5 g C$_3$N$_4$ powder was exfoliated in deionized water (400 ml) for 8 h with a probe ultrasonication cleaner (200 W, UP400S). The dispersion was then centrifuged at 2029 × $g$ for 20 min, yielding the C$_3$N$_4$ photocatalyst. TiO$_2$ was synthesized by the nonhydrolytic sol–gel approach described as follows. A solution of TiCl$_4$ (1 ml), ethanol (5 ml), and benzyl alcohol (35 ml) was incubated for 6 h at 80 °C, then washed three times with diethyl ether. After centrifuging the crude product at 2415 × $g$ for 10 min, a white TiO$_2$ precipitate was obtained. The final TiO$_2$ solution was prepared by dispersing the precipitate in ethanol.

For the CuS–MoS$_2$ synthesis, the process was divided into two steps. A Cu–Mo-based metal-organic framework (i.e., NENU-5) was first prepared through a wet chemical method. 0.6 g copper (II) acetate monohydrate and 1.2 g phosphomolybdic acid hydrate were mixed and sonicated in 40 mL DI water for 30 min. 0.62 g trimeric acid, which was dissolved in 40 ml ethanol, was poured into the above solution quickly, and the nanocrystal NENU-5 was obtained. Second, 2 g sulfur powder and 0.1 g NENU-5 were placed in a dual-zone tube furnace upstream region (250 °C) and down-stream region (550 °C), respectively. The CuS–MoS$_2$ heterojunction was prepared after 1 h reaction under Ar gas environment.

**Synthesis of wood/photocatalyst systems**. Pinewood blocks were cut into pieces using a sweep saw (area: 7.85 cm$^2$, thickness: 5 mm). The whole carbonized wood was directly obtained through heating the woodblock in a muffle furnace at a temperature of 300 °C for 2 h. To improve the steam generation, the surface of the wood slice was treated by a simple heating process to obtain the surface carbonized wood. In detail, the wood samples were pretreated in an alcohol flame for 2 min, then immediately immersed in cold water at room temperature for rapid quenching. Next, the NP solutions (0.3 g of CoO, MoS$_2$, C$_3$N$_4$, TiO$_2$, or CuS–MoS$_2$) were spin-coated onto the wood surface at 500 rpm for 20 s. Finally, the samples were dried in an oven at 45 °C for 2 h, yielding the wood/catalyst systems. The different CoO loadings (0.1, 0.2, 0.3, and 0.4 g) were realized by changing the photocatalyst solution concentration.

**Characterization of the photocatalysts**. The morphologies of the samples were characterized by a scanning electron microscope (SEM) (FEI NOVASEM) and a transmission electron microscope (TEM) (FEI Talos F200X). The infrared radiation thermal image from the wood/CoO system under light illumination was recorded with a UTi80 thermal imager. XPS spectra of wood/CoO and element S for CuS/MoS$_2$ were collected through Kratos Axis supra XPS spectrometer.The local temperature of CoO NPs was estimated through the atomic force microscopy (AFM) (Bruker Dimension Icon). At the 300 and 308 K, the potentials of wood/CoO were measured. After that, the potential of wood/CoO was recorded under

100 mW cm$^{-2}$ light illumination. Due to the correlation between potential and the temperature[21], the local temperature of CoO under light illumination could be estimated. The bulk water of wood/CoO was measured through Perkin Elmer Lambda 35 ultraviolet–visible spectrophotometer and Thermo Fisher ICAP7600-DUO inductively coupled plasma emission spectrometer.

**Solar-driven-steam generation measurements**. Steam generation from the wood was measured in a quartz beaker containing deionized (DI) water. The solar light was simulated by a solar simulator at AM 1.5 G illumination (100 mW cm$^{-2}$), and the mass changes in the water were measured by a high-accuracy balance (Mettler–Toledo, ME204E). The quartz beaker containing the wood in DI water was placed on the balance, and the weight loss of the water was recorded by reading the balance every 3 min under light illumination.

The solar-to-steam-conversion efficiency $\eta$ was calculated as:

$$\eta = \frac{\Delta m * \Delta vap * H}{M * P * S * T} \qquad (4)$$

where $\Delta m$ is the mass loss of water during irradiation, $\Delta vap*H$ is the phase change enthalpy of water from liquid to vapor which is ~40.637 kJ mol$^{-1}$, $M$ is the molar mass of water, $P$ is the solar power density (100 mW cm$^{-2}$), $S$ is the area (about 7.85 cm$^2$), and $T$ is the irradiation time (3600 s).

**Hydrogen generation tests**. For the hydrogen evolution measurement in the liquid water/photocatalysts/hydrogen-gas triphase system, 50 ml of deionized water was added to the transparent reactor chamber, and then the filter paper with photocatalyst was immersed in the water. The reaction cell was placed 7.0 cm from the light source. The light source was a solar simulator at AM 1.5 G illumination (100 mW cm$^{-2}$) (CEL-NP2000) (Supplementary Fig. 32), which was equipped with a fan that efficiently dissipated the excess heat. The reaction temperature in the quartz cell was tuned through a heating jacket and was measured by a thermometer. During the photocatalytic reaction, the gases were transferred into the sample loop by a peristaltic pump and were further quantified by gas chromatography (Shimadzu GC-2014c; Ar carrier gas and molecular sieve-5A column), equipped with a thermal conductivity detector. The hydrogen-gas yield of the reactor was measured every 15 min.

The hydrogen evolution in the injected water steam/photocatalysts biphase system was carried out similarly to the triphase reaction measurement. The filter paper with photocatalysts was taken to the middle of the quartz cell. The steam was injected into the quartz cell, and the steam flowmeter was used to monitor the steam flux. The remaining testing process in the biphase system was similar to that of the triphase system evolution.

The AQY is calculated based on the formula below[60]:

$$AQY = \frac{2 * n * N_A}{(E * A * T * \lambda)/(h * C)} \times 100\% \qquad (5)$$

where $n$ is the H$_2$ yield, $N_A$ is the Avogadro number, $E$ is light intensity, $A$ is the irradiation area, $T$ is the time, $\lambda$ is the wavelength, $h$ is the Planks constant, and $C$ is the speed of light.

The H$_2$O$_2$ concentration was determined through UV–Vis absorption spectra[61]. 0.01 mol L$^{-1}$ copper (II) sulfate solution was prepared in advance, and 1 g neocuproine was dissolved in 100 ml ethanol to obtain 2,9-dimethyl-1,10-phenanthroline (DMP) solution. 5 μmol ml$^{-1}$ H$_2$O$_2$ was used as a standard sample to react with copper (II) sulfate solution and DMP solution. The obtained absorption spectrum was the baseline. Then, the solutions after photocatalytic reaction were mixed with copper (II) sulfate solution and DMP solution to measure related absorption spectra, which could be utilized to calculate the H$_2$O$_2$ concentration through comparing with the baseline.

The hydrogen evolution measurement in the wood/photocatalyst photothermal–photocatalytic system was carried out similarly to the triphase reaction measurement. The quartz cell contained 50 ml of deionized water. The wood/photocatalyst systems were floating on the water. The remaining testing process was similar to that of the triphase system hydrogen evolution measurement. In simulated seawater splitting to hydrogen measurement, the simulated seawater contains 3.1 wt% NaCl, 0.2 wt% MgCl$_2$, and 0.1 wt% KCl; and the wood/photocatalyst systems (0.3 g CuS–MoS$_2$ were loaded) were floating on the simulated seawater. After the photocatalytic reaction, the hydrogen collector was taken to be measured through gas chromatography. The hydrogen collector was replaced by a new one every 2 h to carry out the photocatalytic stability measurement.

**Theoretical calculation**. All periodic calculations were performed in the Vienna Ab Initio Simulation Package (VASP), which was on the basis of the generalized gradient approximation (GGA) and the exchange-correlation energy of interacting electrons determined by the revised-Perdew–Burke–Ernzerhof (RPBE) functional. The ion–electron interaction was described with the projector augmented wave (PAW) method[62]. A basis set of plane waves was up to an energy cutoff of 520 eV. The CoO (111) surface was modeled with a 2 × 2 supercell containing 13 atomic layers, where 5 layers were fixed in the bulk positions. All slab structure included a vacuum of 15 Å. The dipole moment correction was considered and added in calculation optimization process. And the antiferromagnetic moment was set up

along the (111) direction. A U value of 4.1 eV was applied to the Co d-states. The Monkhorst-pack method with the centered k-point grid ($4 \times 4 \times 1$) was used for surface calculations, respectively. The convergence threshold for the residual force was set to 0.02 eVÅ$^{-1}$, and energies have converged within $10^{-5}$ eV. The model structure schemes and band structures of CoO have been shown in the Supplementary Figs. 33 and 34.

The hydrogen and water adsorption energy on various surfaces is defined as[63,64]

$$\Delta E_{ads} = E_{base-H} - E_{base} - \frac{1}{2} E_{H_2} \qquad (6)$$

$$\Delta E_{ads} = E_{base-H_2O} - E_{base} - E_{H_2O} \qquad (7)$$

where $E_{base-H}$ and $E_{base-H_2O}$ are the total energy of the slab model with H and H$_2$O adsorption, $E_{base}$ is the energy of a clean slab surface, and $E_{H_2}$ and $E_{H_2O}$ are that for hydrogen and water molecules.

The Gibbs energy can be calculated by taking zero-point energy and entropy corrections into account[65] such that $\Delta G = \Delta E + \Delta E_{ZPE} - T\Delta S + \Delta G_{pH}$. Where $\Delta E_{ZPE}$ and T$\Delta S$ are the difference in zero-point energy and entropy between the adsorbed species and free species in the gas phase, respectively[66,67]. At different pH values, $\Delta G_{pH} = 0.059 \times pH$. The solvent effect is considered through the implicit solvation model based on the VASPsol[68,69]. The dielectric constants of liquid water and gas water are indexed to be 81 and 1, respectively. The differences of Gibbs free energies in the bi- and tri-phasic systems are the temperature (373 and 298 K, respectively) and the entropy change $\Delta S$. The dielectric constant and entropy change $\Delta S$ correction in the vapor water and liquid water was obtained from the Handbook of Chemistry and Physics[70]. The entropy $S$ is 0.367 kJ kg$^{-1}$ K$^{-1}$ at 298 K under the standard pressure, while at 373 K the entropy $S$ is 1.303 kJ kg$^{-1}$ K$^{-1}$ when the water is liquid (the value is used for Gibbs free energies calculation in tri-phasic systems at 373 K) and 7.361 kJ kg$^{-1}$ K$^{-1}$ when the water is in the gas phase (the value is used for Gibbs free energies calculation in biphasic systems at 373 K).

## Data availability

The authors declare that the data supporting the findings of this study are available within this paper and its Supplementary information file, or from the corresponding authors.

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

## Acknowledgements

This research is supported by the Basic Research Fund for Free Exploration in Shenzhen (Grant No. JCYJ20180306171402878), the project of Shaanxi Young Stars in Science and Technology (2017KJXX-18), the Shaanxi International Cooperation Project (2020KWZ-018), and the Fundamental Research Funds for the Central Universities (3102019ghxm003, 3102019JC005, 3102019ghjd001). We thank the members from the Analytical & Testing Center of Northwestern Polytechnical University for the help of TEM and XPS characterization. We also thank Prof. Weihong Qi, Prof. Qingfeng Zeng, and Prof. Junjie Wang for valuable discussion about the theoretical calculation.

## Author contributions

B.W. and X.L. conceived the concept and directed the research. S.G. and X.L. designed the project. S.G. carried out material synthesis and related characterization tests. J.L. gave advice on the experiments. S.G., X.L., and B.W. wrote the paper. All authors discussed the results and commented on the paper.

## Competing interests

The authors declare no competing interests.
