## [Peer Review File · Nature Communications]

Reviewers' comments:

Reviewer #1 (Remarks to the Author):

In this work, the author proposed a new strategy to advance the photocatalytic performance of photocatalysts for the hydrogen production from water, in which solar energy is used to evaporate the water to produce vapor and H₂ can be produced by the photocatalytic reaction of vapor. The highly efficient hydrogen production should be attributed to the reduced interface barrier in the adsorption and transport resistance in this biphasic system with respect to the case that catalytic reaction is performed in bulk water. This work shows some potential in the advance of hydrogen production from water using solar energy. Here are some comments:

- 1) Since the adsorption of vapor is of significance to the production of hydrogen and this process is likely to be governed by the flow rate of the steam. Therefore, it is essential to add some discussions on the relation between the steam flow rate and hydrogen production rate.
- 2) In the catalysis process, H₂O₂ was produced from water, and these H₂O₂ is able to be accumulated on the surface of the wood. Will these accumulated H₂O₂ have any effect on the durability of the device when it was maintained under illumination for a long period (e.g. 8 h for 5 days)?
- 3) The catalyst was spin coated on the surface of the wood, so will the catalyst be very easy to detach from the substrate? This might cause the loss of the catalyst, resulting in the contamination of bulk water.
- 4) The catalytic reactivity of the photocatalytic reaction is related to the solar intensity and reaction temperature. Therefore, the solar intensity might not be in a linear relation with the hydrogen production rate. Their relationship need to be further explored.

Reviewer #2 (Remarks to the Author):

The manuscript describes an experimental study on the photothermal-photocatalytic properties of wood/CoO composite materials. The authors extensively discuss the hydrogen evolution rates of the prepared material, the influence of the state of water (liquid vs. gas) and try to reveal the universality of the approach by utilization of various light-absorbing materials. As such applying photothermal transpiration to produce (solar) steam without additional energy input is quite interesting but the reported data raise some questions. Particularly, the absence of stoichiometric amounts of oxygen (or hydrogen peroxide) are indicative for either a sacrificial reagent (e.g. the wood) participating in the reaction or the general instability of the CoO and particularly the CuS/MoS₂ catalyst where it is even highlighted that NO₂/H₂O₂ has been detected; so what is oxidized in this case? Therefore, claiming overall water splitting is not justified and any comparison with other systems performing overall water splitting is meaningless. Additionally, it is widely accepted in photocatalysis that a comparison photocatalysts based on production rates should be avoided. Instead QY or AQY should be reported. It is also advised to report rates here on a per cm² basis rather than a per g basis given the significant dimensions of the wood used. Generally, it is advised to thoroughly reconsider the wording, i.e. record-high hydrogen production. Additional minor details are:

- The authors suggest stability of the system based on (HR)TEM data. However, comparing Fig S5 and S14 significant agglomeration is evident and the surface of the particles seems to be less ordered. Given that the wood substrate is essential a thorough characterization is required. This should be included and stability of the system clarified.
- It is not clear whether only steam is reacting to H₂ or also liquid water is involved in the reaction. Given that hydrogen peroxide is detected in the liquid phase it might be the later. The author should provide a concise picture illustration the processes occurring.
- The dependence of CoO loading should be addressed. Is this this an optimized configuration?
- The authors report that a solar simulator is used placed 7 cm above the sample. However, it is not

clear whether 100 mW/cm² are obtained at the sample surface.

- The authors report thermal images and indicate that only a minor increase in temperature is observed at the surface of the wood/CoO system. The local temperature at the nm scale (size of the CoO particles) however might be much higher. The authors should comment on local vs. global temperature effects.

- The authors should use appropriate and original literature, i.e. techno-economics used to justify the benefits of a PC system are not reported in Ref. 2. Additionally it is questionable whether the techno-economics of a PC system are actually applicable in this case as the reactor design and dimensions are significantly deviating from a true PC system.

Given the above I would recommend NOT to publish the manuscript in Nature Communications. Major revisions and a significant modification, including discussion of additional data concerning stability, is required before resubmission and additional review.

Reviewer #3 (Remarks to the Author):

In the present manuscript the authors add a novel twist to the by now well known class of carbon-supported nanoparticle catalysts where, instead of using a well defined carbon material, they use charred wood. While this unquestionably makes the whole setup much less well defined than even mesoporous carbon supports, they were able to show that this setup leads to tremendously improved hydrogen production rates in photo-catalytic water splitting. The effect of the charred wood thereby seems to be the fact that it produces water vapour near the catalytic nanoparticles which then favourably influences the overall reaction rates. Given that humanity is still on the hunt for adequate alternative energy production methods and storage media, the discovery of a novel photocatalyst for water splitting is in principle of utmost importance. All the more so considering the ubiquity of the new support material.

Due to my background in theory I do not think my opinion on the experimental methods used in this work will be necessary. I will, however, comment on the theoretical calculations the authors conducted, which, frankly, are so far below the current state of the art to render the results essentially meaningless. As I will outline in detail below, I do not think that any of the calculations undertaken here are of sufficient predictive quality, which to be honest also would make me somewhat suspicious of the rest of the work.

These are the specific problems I found with the manuscript:

- 1) The structural model. Which slab did the authors use, was it a surface unitcell or a supercell, how many layers did they include? This is really the most basic information and the authors fail to give it.
- 2) I assume, because it is never actually stated, that the authors simulated CoO in its rocksalt structure. In which case the O-terminated (111) facet is actually partially charged, meaning their simulation cell should (if they simulated stoichiometric CoO, also not stated) show a quite sizeable dipole perpendicular to the surface. This is well known and most works compensate for this, yet there is no mention of such a compensation in this work.
- 3) One possibility why the authors did not encounter a surface dipole is because the PBE functional used in this work likely yields a metallic electronic structure for CoO. Not knowing the specifics, I would nevertheless assume that their material has a band gap, otherwise it would not be a photocatalyst. Again, this is a well known and understood failure of GGA functionals. Yet, with such a broken electronic structure I do not think that the calculated binding energies show anything but a fortuitous overlap with reality.
- 4) Why did the authors simulate the de-protonated CoO facet? What pH are they aiming for? Most studies nowadays at least try to determine the protonation state of their surface (which btw. could be one way to avoid the aforementioned surface dipole) beforehand.

5) It is completely unclear to me how the authors are supposed to have calculated the Gibbs free energies in the bi- and tri-phasic systems. Did they actually include solvent effects? If so, how? As it is, the way Figure 3c came to be is entirely unclear to me.

6) In order to determine some of the Gibbs free energies, the authors apply a technique now known as the computational hydrogen electrode, by J. Norskov and J. Rossmeisl (eq. 5 in the present work). While this is indeed state of the art, I find it mildly suspicious that the authors neglect to cite any of the original works here.

7) Finally, not a point directly related to the first-principles calculations, but with eq. 1 the authors calculate the effective free energy barrier of their reaction, yet fail to use it for anything than mildly extrapolating towards higher temperatures. Why is this barrier not discussed or at least compared to other setups? I am sure the kinetic barriers on pristine CoO can by now be found in the literature.

These 7 points together paint a very grim picture of at least the theoretical parts of this manuscript. I thus cannot recommend its publication in Nature Communications, or indeed anywhere else.

Point-by-point Response to the Reviewers' Comments

Reviewer #1

In this work, the author proposed a new strategy to advance the photocatalytic performance of photocatalysts for the hydrogen production from water, in which solar energy is used to evaporate the water to produce vapor and H₂ can be produced by the photocatalytic reaction of vapor. The highly efficient hydrogen production should be attributed to the reduced interface barrier in the adsorption and transport resistance in this biphasic system with respect to the case that catalytic reaction is performed in bulk water. This work shows some potential in the advance of hydrogen production from water using solar energy. Here are some comments:

1) Since the adsorption of vapor is of significance to the production of hydrogen and this process is likely to be governed by the flow rate of the steam. Therefore, it is essential to add some discussions on the relation between the steam flow rate and hydrogen production rate.

Response: Thanks very much for the valuable suggestion. The hydrogen production rate as a function of flow rate the water steam (from 5 ml/h to 88 ml/h), as shown in Figure R1, can be easily calculated from the photocatalytic hydrogen gas yield in Figure R2, which was provided as Figure S13 in the original submission. The rate of hydrogen production from steam increases along with the increase of steam flow rate from 5 ml/h to 62 ml/h and is then stabilized when the steam flow rate is further increased. This is because more water molecules are needed to participate in the

hydrogen evolution reaction until saturation is reached, i.e., the optimal steam flow rate is about 62 ml/h.

In the revised version, this discussion has been added in the main text; and Figure R 1 is added as Supplementary Figure 18 in the Supporting Information.

Figure R1. The hydrogen production rate from the water steam with different flow rates.

Figure R2 (*i.e.*, Supplementary Figure 17 in the original version). Time-dependent photocatalytic hydrogen gas production profiles from the liquid water and water steam with different flow rates. The photocatalyst is CoO NPs.

2) In the catalysis process, H₂O₂ was produced from water, and these H₂O₂ is able to be accumulated on the surface of the wood. Will these accumulated H₂O₂ have any

effect on the durability of the device when it was maintained under illumination for a long period (e.g. 8 h for 5 days)?

Response: We thank the reviewer for the insightful comment. According to the reviewer's suggestion, we have carried out the durability measurement on the wood/CoO system for an extended period (8 h for 5 days). As shown in Figure R3, the amount of H₂ evolution increases linearly along with the measurement time (8 h/day for 5 days). On day 5, the photocatalytic hydrogen evolution performance maintains about 90 % of that in day 1, indicating the good durability of the wood/CoO device. The accumulated H₂O₂ seems to have little effect on hydrogen production stability.

In the revised version, Figure R3 is added as a new Figure 2d; and the related discussion has been added in the main text.

Figure R3. Time-dependent photocatalytic gas production profiles from the wood/CoO for a long-period measurement (8 h for 5 days).

3) The catalyst was spin coated on the surface of the wood, so will the catalyst be very easy to detach from the substrate? This might cause the loss of the catalyst, resulting

in the contamination of bulk water.

Response: Thanks very much for the valuable comments. In our work, the catalyst was spin-coated on the surface of the wood, which was dried in an oven at 45 °C for 2 h to help improve the catalyst adhesion to the wood channels. We compared the SEM images and EDS element mapping of CoO NPs attached to the wood microchannels before and after the photocatalytic reaction, as shown in Figure R4 and R5. No apparent changes could be observed. Besides, the UV-Vis spectrum of the water in the wood/CoO system after the photocatalytic reaction has been measured as shown in Figure R6. There are no CoO characteristic peaks in the absorption spectrum, indicating that CoO was not detached from the wood to contaminate the water system.

In the revised version, Figure R6 of the absorption spectrum from the water in the wood/CoO system is added as Supplementary Figure 13 in the Supporting Information; and the related discussion has been added in the main text.

Figure R4 (*i.e.*, Supplementary Figure 10 in the original version). SEM images and EDS element mapping of CoO NPs attached to the walls of the wood microchannels before photocatalytic reaction.

Figure R5 (*i.e.*, Supplementary Figure 11 in the original version). SEM images and

EDS element mapping of CoO attached to the walls of the wood microchannels after the photocatalytic reaction.

Figure R6. The absorption spectrum from the water in the wood/CoO system.

4) The catalytic reactivity of the photocatalytic reaction is related to the solar intensity and reaction temperature. Therefore, the solar intensity might not be in a linear relation with the hydrogen production rate. Their relationship need to be further explored.

Response: Thanks for the great suggestion. Accordingly, solar intensity (100, 200, and 300 mW/cm²) on the photocatalytic reaction of the wood/CoO system has been investigated. As shown in Figure R7, the rate of hydrogen evolution grows with solar intensity, but not linearly related to the hydrogen production rate as predicted by the reviewer. This is mainly because of the rise of temperature on the wood/CoO surface caused by the increase in solar intensity. As shown in Figure R8, the temperature of the wood/CoO surface is increased to 324, 342, and 357 K when the solar power of 100, 200, and 300 mW/cm² is applied, respectively. In addition, we have already studied the temperature effect on the rate of hydrogen evolution under the same solar intensity illumination (100 mW/cm²). As shown in Figure R9, the temperature can exponentially improve hydrogen production. Thus, the relationship between the rate

of hydrogen evolution and solar intensity could be nonlinear.

In the revised version, the rate of H₂ evolution against solar intensity in the wood/CoO system (Figure R7) and infrared radiation thermal images (Figure R8) are added as the new Figure 2c and Supplementary Figure 9 in the Supporting Information, respectively. The related discussion has been added to the main text.

Figure R7. Rate of H₂ evolution versus solar intensity in the wood/CoO system.

Figure R8. Infrared radiation thermal images of the wood/CoO system under light illumination with different solar intensity. **a** 100 mW/cm², **b** 200 mW/cm², **c** 300 mW/cm².

Figure R9 (*i.e.*, Figure 4a in the original version). The photocatalytic hydrogen evolution rate with the CoO NPs versus the reaction temperature in the triphase reaction system.

Reviewer #2:

The manuscript describes an experimental study on the photothermal-photocatalytic properties of wood/CoO composite materials. The authors extensively discuss the hydrogen evolution rates of the prepared material, the influence of the state of water (liquid vs. gas) and try to reveal the universality of the approach by utilization of various light-absorbing materials. As such applying photothermal transpiration to produce (solar) steam without additional energy input is quite interesting but the reported data raise some questions. Particularly, the absence of stoichiometric amounts of oxygen (or hydrogen peroxide) are indicative for either a sacrificial reagent (e.g. the wood) participating in the reaction or the general instability of the CoO and particularly the CuS/MoS₂ catalyst where it is even highlighted that NO O₂/H₂O₂ has been detected; so what is oxidized in this case? Therefore, claiming overall water splitting is not justified and any comparison with other systems performing overall water splitting is meaningless.

Response: Thanks for the comments. The typical photocatalysts we selected in this work mainly produce H₂, and only the half-reaction for H₂ production has been involved. There are three reaction paths for the photo-induced holes in the photocatalytic hydrogen production reaction, including (1) reaction with sacrificial reagents, (2) by-products formation such as H₂O₂, and (3) reaction with the catalyst surface. In the wood/CoO case, the main reaction path for the photo-induced holes is H₂O₂ production, which has been confirmed by detecting the quantity of H₂O₂ after the photocatalytic reaction. As shown in Figure RR1, the H to O ratio would satisfy

2:1 with the consideration of the H_2O_2 amount. This is convincing evidence to prove the wood without the sacrificial agent effect. In the CuS/MoS_2 case, H_2O_2 is not detectable. Some previous studies demonstrated the surface reaction for metal sulfides during the photocatalytic reaction process (J. Mater. Chem. A, 2015, 3, 13913; Sol. Energy, 2018, 171, 106; Catal. Commun., 2014, 44, 62). Similarly, the photo-induced holes could react with S ions from the CuS/MoS_2 catalysts in the current work as shown based on the XPS results (Figure RR2).

Considering the oxygen generation issue, we only compared photocatalytic hydrogen production performance with other systems rather than the overall water splitting. Therefore, the comparison is reasonably fair.

Figure RR1 (*i.e.*, Supplementary Figure 16 in the original version). The H_2O_2 concentration determination from absorption spectra. (a) the absorption spectra from different H_2O_2 concentrations and (b) the linear fitting between the intensity of absorption peak and H_2O_2 concentration.

Figure RR2 The high-resolution XPS of element S for CuS/MoS₂ before and after photocatalytic reaction.

Additionally, it is widely expected in photocatalysis that a comparison photocatalysts based on production rates should be avoided. Instead QY or AQY should be reported. It is also advised to report rates here on a per cm² basis rather than a per g basis given the significant dimensions of the wood used.

Response: Thanks for the comments. It is understood that QY or AQY, determined by the photoelectric properties of a photocatalyst, should be provided if a new catalyst is developed. However, in the current work, we focused on the interface regulation between water molecules and photocatalysts rather than the photocatalysts themselves. The photocatalysts we selected to demonstrate our proof-of-concept are all typical photocatalysts and have been intensively reported in the literature. In the wood/photocatalyst system, only the catalytic environment is varied from liquid water to water steam, and nothing has changed regarding the photocatalysts. Therefore, it is not necessary to measure the QY or AQY here.

We thank the reviewer for the excellent advice on the unit, i.e., the production rate per cm^2 . In the revised version, we modified the production rate quantitative unit in Figures 2 (a)-(c).

Generally, it is advise to thoroughly reconsider the wording, i.e. record-high hydrogen production.

Response: We agree with the reviewer and modified “record-high hydrogen production” to be “impressive hydrogen production”.

Additional minor details are:

- The authors suggest stability of the system based on (HR)TEM data. However, comparing Fig S5 and S14 significant agglomeration is evident and the surface of the particles seems to be less ordered. Given that the wood substrate is essential a thorough characterization is required. This should be included and stability of the system clarified.

Response: Thanks very much for the valuable comments. First, the catalyst CoO NPs were spin-coated to the wood matrix, and it is challenging to realize the uniform dispersion of CoO NPs. However, the current preparation method is straightforward, low-cost, and greatly advantageous for practical applications. In the future, some templated methods could be considered for ordered NPs loading. Regarding the thorough characterization of the wood/CoO, we further performed the TEM and EDS element mapping to investigate the whole wood/CoO system. As shown in Figures RR3 and RR4, the CoO NPs are still attached to the wood channels after the

photocatalytic reaction, exhibiting the good adhesion stability of photocatalysts. In addition, the UV-Vis spectrum of the bulk water in the wood/CoO system after the photocatalytic reaction has been measured as shown in Figure RR5. There is no CoO characteristic peak in the absorption spectrum, exhibiting that the catalyst CoO is still attached to the wood. Moreover, the reflection spectra of the wood/CoO system have also been measured before and after the photocatalysis process, as shown in Figure RR6. There is little difference in the reflection spectra of the wood/CoO system before and after the photocatalysis process, exhibiting the excellent stability of the wood/CoO system.

In the revised version, the TEM images (Figure RR3 and RR4), reflection spectra (Figure RR6) of the wood/CoO system before and after the photocatalytic reaction, and the absorption spectrum of the bulk water in the wood/CoO system after the photocatalytic reaction (Figure RR5) are added as the new Figures 2e and 2f in the main text and as Supplementary Figures 12 and 13 in the Supporting Information. Related discussions have been added to the main text.

Figure RR3. TEM image and EDS element mapping of CoO NPs attached to the walls of the wood microchannels before photocatalytic reaction.

Figure RR4. TEM image and EDS element mapping of CoO attached to the walls of the wood microchannels after the photocatalytic reaction.

Figure RR5. The absorption spectrum from bulk water in the wood/CoO system after the photocatalytic reaction.

Figure RR6. The reflection spectra of the wood/CoO system before and after the photocatalysis process.

- It is not clear whether only steam is reacting to H_2 or also liquid water is involved in the reaction. Given that hydrogen peroxide is detected in the liquid phase it might be the later. The author should provide a concise picture illustration the processes occurring.

Response: Thanks very much for the valuable comments. In the current work, only

steam participated in producing H_2 . Liquid water was not involved in the reaction because the photocatalysts in the wood system did not have contact with liquid water. This claim is evidenced by the Raman spectrum to confirm the CoO NPs distributed approximately 2 mm along the walls of the wood microchannels. As shown in Figure RR7, when the wood/CoO system floats on the surface of the water, the immersion depth of the wood in the water is about 2 mm, indicating that the photocatalysts are not directly soaked in the liquid phase water.

It is unfortunate that “hydrogen peroxide is detected in the liquid water” generated confusion. The reality is H_2O_2 was produced from the photocatalytic reaction with steam in the wood/CoO system. To analyze the H_2O_2 quantity with absorption spectra (Figure RR1), we had to wash the wood with liquid water, into which H_2O_2 was fully dissolved.

To eliminate the misunderstanding, we redraw a schematic of the photocatalytic reaction process, as shown in Figure RR8. Under light illumination, the adhered photocatalysts become covered with water steam produced by the photothermal transpiration in the wood interior. Simultaneously, the photo-induced electrons generated in the photocatalysts participate in the hydrogen evolution reaction at the photocatalytic active sites, and the photo-induced holes participate in the H_2O_2 generation. The new schematic of the photocatalytic reaction process (Figure RR8) is added as Figure 1a, and the related discussion has been added in the revised manuscript.

Figure RR7 (*i.e.*, Figure 1 in the original version). (left) Raman spectra taken at different depths along the cross-section of the wood/CoO microchannels with an interval of 500 μm . (right) the photo of the wood/CoO system floating on the surface of water.

Figure RR8. Schematic of the photocatalytic reaction process of the wood/CoO system.

- The dependence of CoO loading should be addressed. Is this an optimized configuration?

Response: Thanks very much for the valuable suggestion. We investigated the effect of CoO NPs mass loading in the wood/CoO system on the photocatalytic hydrogen gas production rates (see Figure RR9) via varying the CoO concentration during the

spin-coating process. Photocatalytic hydrogen production rates with different catalyst mass loadings are analyzed within a constant surface area of wood (area: 7.85 cm^2). An optimized mass loading of about 38 mg/cm^2 CoO NPs has been identified based on the experimental results.

In the revised version, Figure RR9 is added as the new Figure 2a in the main text; and the related discussion has been added in the revised manuscript.

Figure RR9. Mass loading-dependent photocatalytic hydrogen gas production rates for the wood/CoO system (area: 7.85 cm^2).

- The authors report that a solar simulator is used placed 7 cm above the sample. However, it is not clear whether 100 mW/cm^2 are obtained at the sample surface.

Response: Thanks for raising the question. In fact, the light intensity of 100 mW/cm^2 was calibrated at the distance of 7 cm from the solar simulator through the optical power meter (CEL-NP2000), as shown in Figure RR10.

In the revised version, the photograph of the light intensity measurement is added as Supplementary Figure 33 in the supporting information.

Figure RR10. The photograph of the light intensity measurement.

- The authors report thermal images and indicate that only a minor increase in temperature is observed at the surface of the wood/CoO system. The local temperature at the nm scale (size of the CoO particles) however might be much higher. The authors should comment on local vs. global temperature effects.

Response: Thanks very much for the insightful comments. We agree with the reviewer that the local temperature could be significantly different from the global temperature. The local temperature of the nanoscale CoO particles under light illumination can be estimated through the potential, which can be measured using Atomic Force Microscopy (Annu. Rev. Mater. Sci., 1999, 29, 505). According to this reference, there is a correlation between the potential and the temperature. Therefore, we first measured the potential of CoO particles at two different temperatures (the temperature was pre-set), 300 K and 308 K, as shown in Figures RR11 a and b. Then, the potential of CoO particles under 100 mW/cm^2 light illumination was measured, as shown in Figure RR11 c. Based on the relationship between potential and the temperature, the local temperature of CoO particles under 100 mW/cm^2 light

illumination is estimated to be 346 K (Figure RR11 d), which, as expected, is higher than the global temperature (325 K, in Figure RR12) because of the nanoscale effect. It is speculated that a higher local temperature is beneficial to enhance the photocatalytic reaction efficiency.

In the revised version, Figure RR11 is added as Figures 1f and 1g in the main text and Supplementary Figure 7 in the Supporting Information; and the related discussion has been added in the main text. We thank the reviewer for a very useful suggestion.

Figure RR11. **a** The potential of wood/CoO at 300 K, **b** the potential of wood/CoO at 308 K, **c** the potential of wood/CoO under 100 mW/cm² light illumination, and **d** the estimated temperature through the potential.

Figure RR12 (*i.e.*, Figure 1e in the original version). Infrared radiation thermal image from the wood/CoO system under light illumination.

- The authors should use appropriate and original literature, *i.e.* techno-economics used to justify the benefits of a PC system are not reported in Ref. 2. Additionally it is questionable whether the techno-economics of a PC system are actually applicable in this case as the reactor design and dimensions are significantly deviating from a true PC system.

Response: Thanks for the comments. We have double checked and confirmed that the “techno-economics used to justify the benefits of a PC system” was expressed in the first paragraph of Ref. 2 though this techno-economic discussion of photocatalysis was originally reported in the literature (Energy Environ. Sci., 2013, 6, 1983). Anyway, to avoid confusion and downplay the techno-economic discussion, we have rephrased the sentence: “There are three main types of solar-driven hydrogen production systems: particulate photocatalysis, photovoltaic-assisted electrolysis (PV-E), and photoelectrochemical cells (PEC)², where the particulate photocatalysis is predicted to be more cost-effective than the other two systems⁴”, in the revision. The original literature (Energy Environ. Sci., 2013, 6, 1983) was also cited as ref. 4.

Reviewer #3:

In the present manuscript the authors add a novel twist to the by now well known class of carbon-supported nanoparticle catalysts where, instead of using a well defined carbon material, they use charred wood. While this unquestionably makes the whole setup much less well defined than even mesoporous carbon supports, they were able to show that this setup leads to tremendously improved hydrogen production rates in photo-catalytic water splitting. The effect of the charred wood thereby seems to be the fact that it produces water vapour near the catalytic nanoparticles which then favourably influences the overall reaction rates. Given that humanity is still on the hunt for adequate alternative energy production methods and storage media, the discovery of a novel photocatalyst for water splitting is in principle of utmost importance. All the more so considering the ubiquity of the new support material.

Due to my background in theory I do not think my opinion on the experimental methods used in this work will be necessary. I will, however, comment on the theoretical calculations the authors conducted, which, frankly, are so far below the current state of the art to render the results essentially meaningless. As I will outline in detail below, I do not think that any of the calculations undertaken here are of sufficient predictive quality, which to be honest also would make me somewhat suspicious of the rest of the work.

Response: Thanks very much for the valuable comments. In the revised version, we have tried our best to strengthen the theoretical calculation with more direct evidences and sufficient details. The new results still support our conclusion. The retained results

will be discussed with specific problems the reviewer asked below.

These are the specific problems I found with the manuscript:

1) The structural model. Which slab did the authors use, was it a surface unitcell or a supercell, how many layers did they include? This is really the most basic information and the authors fail to give it.

Response: Thanks very much for the valuable comments. As shown in Figure RRR1, The (111) surface of CoO with rock salt crystal structure was modeled with a 2×2 supercell containing 13 atomic layers, where 5 layers were fixed in the bulk positions. All slab structure included a vacuum of 15 Å. Besides, H and OH groups were also modeled to the CoO (111) surface to comparatively analyze the photocatalytic process.

In the revised version, Figure RRR1 of structure schemes of CoO, CoO-H, and CoO-OH has been added as Supplemental Figure 34 in the Supporting Information; and the details of the CoO model structures have been added in the Theoretical Calculation section of the main text.

Figure RRR1. The model structure schemes of (a) CoO, (b) CoO-H, and (c)

CoO-OH.

2) I assume, because it is never actually stated, that the authors simulated CoO in its rocksalt structure. In which case the O-terminated (111) facet is actually partially charged, meaning their simulation cell should (if they simulated stoichiometric CoO, also not stated) show a quite sizeable dipole perpendicular to the surface. This is well known and most works compensate for this, yet there is no mention of such a compensation in this work.

Response: Thanks very much for the valuable comments. In our calculation, the dipole compensation was applied to eliminate the field influence from the asymmetric plate system. In other words, the dipole moment correction was considered and added in the calculation optimization process. The antiferromagnetic moment was set up along the (111) direction.

In the revised version, we have added the crystal structure of CoO and the clarification of the dipole compensation in the Theoretical Calculation section.

3) One possibility why the authors did not encounter a surface dipole is because the PBE functional used in this work likely yields a metallic electronic structure for CoO. Not knowing the specifics, I would nevertheless assume that their material has a band gap, otherwise it would not be a photo-catalyst. Again, this is a well known and understood failure of GGA functionals. Yet, with such a broken electronic structure I do not think that the calculated binding energies show anything but a fortuitous overlap with reality.

Response: Thanks very much for the valuable comments. In our work, DFT+U calculations were performed using VASP with PAW pseudopotentials provided in the VASP database and the RPBE-generalized gradient approximation (GGA) exchange-correlation functional. A U value of 4.1 eV was applied to the Co d-states. Based on the methods, the band structures of CoO (including CoO, CoO-H, and CoO-OH) have been obtained. As shown in Figure RRR2, the CoO (including CoO, CoO-H, and CoO-OH) show semiconductor band structures rather than the metallic electronic structure. The bandgap is about 2.3 eV, which is consistent with the experimental results based on the absorption spectrum of CoO NPs (Supplementary Figure 6).

In the revised version, Figure RRR2 of band structures of different CoO models, including CoO, CoO-H, and CoO-OH, have been added as Supplemental Figure 35 in the Supporting Information; and the details of the calculation method have been supplemented in the Theoretical Calculation section.

Figure RRR2. The band structures of CoO (a), CoO-H (b), and CoO-OH (c).

4) Why did the authors simulate the de-protonated CoO facet? What pH are they aiming for? Most studies nowadays at least try to determine the protonation state of their surface (which btw. could be one way to avoid the aforementioned surface dipole)

beforehand.

Response: Thanks very much for the valuable comments. In our work, the surface dipole was considered and corrected in the calculation optimization process (see the answers to Question 2). And the de-protonated CoO facet was simulated to analyze the hydrogen adsorption on the pure CoO surface here (Experimentally, the pH value of the reactant water is approximately equal to 7). Besides, the CoO facet was constructed with the H and OH groups to simulate the different CoO status in the water molecules environment. It was possible that some groups (i.e., H and OH) could be attached to the surface of CoO. The surface groups (i.e., H and OH) could also realize the surface dipole compensation.

Here, the pure CoO structure, CoO with H groups, and CoO with OH groups could simulate the different situations in the neutral, acid, alkaline environment. Moreover, the pH effect was also considered implicitly through the formula $\Delta G = \Delta E + \Delta E_{ZPE} - T\Delta S + \Delta G_{pH} + \Delta G_{sol}$. ΔE is the adsorption energy, ΔE_{ZPE} is the difference in zero point energy, T is the temperature, and ΔS is the difference in entropy between the adsorbed species and free species in the gas phase. At different pH values, $\Delta G_{pH} = 0.059 \times \text{pH}$; and ΔG_{sol} represents the correction terms for solvent effect (0 eV for H* and 0.5 eV for OH*) (J. Mater. Chem. A, 2019, 7, 3648). The Gibbs free energies of hydrogen evolution reaction from the three different CoO structures were calculated to analyze the thermodynamics in the photocatalytic process. As shown in the Figures RRR3-RRR5, in the different pH environment, both temperature and water phase status significantly affect the water molecule adsorption

process to reduce the barriers, which demonstrate advantageous to the photocatalytic hydrogen evolution reaction.

In the revised version, Figure RRR3 of the Gibbs free energies of the hydrogen evolution reaction from the pure CoO structure is added as Figures 4b and 4c, and Figures RRR4 and RRR5 of the Gibbs free energies of the hydrogen evolution reaction from the CoO-H and CoO-OH structures are added as Supplemental Figures 21 and 22 in the supporting information. And the related discussion is added in the main text.

Figure RRR3. **a** Gibbs energy of a photocatalytic reaction on the pure CoO surface at different temperatures in the triphase system. **b** Gibbs energy of a photocatalytic reaction on the pure CoO surface in the triphase system in comparison with the biphasic system at 373 K.

Figure RRR4. **a** Gibbs energy of a photocatalytic reaction over the CoO-H structure at different temperatures in the triphase system. **b** Gibbs energy of a photocatalytic

reaction over the CoO-H structure in the triphase system in comparison with the biphase system at 373 K.

Figure RRR5. **a** Gibbs energy of a photocatalytic reaction over the CoO-OH structure at different temperatures in the triphase system structure. **b** Gibbs energy of a photocatalytic reaction over the CoO-OH structure in the triphase system in comparison with the biphase system at 373 K..

5) It is completely unclear to me how the authors are supposed to have calculated the Gibbs free energies in the bi- and tri-phasic systems. Did they actually include solvent effects? If so, how? As it is, the way Figure 3c came to be is entirely unclear to me.

Response: Thanks very much for the valuable comments. In our work, the Gibbs free energies in both bi- and tri-phasic systems were calculated based on the formula $\Delta G = \Delta E + \Delta E_{ZPE} - T\Delta S + \Delta G_{pH} + \Delta G_{sol}$. ΔE is the adsorption energy. ΔE_{ZPE} is the difference in zero-point energy. T is the temperature, and ΔS is the difference in entropy between the adsorbed species and free species in the gas phase. The differences of Gibbs free energies in the bi- and tri-phasic systems are the temperature (373 K and 298 K, respectively) and the entropy change ΔS . The entropy change ΔS correction in the vapor water and liquid water was obtained from the Handbook of Chemistry and Physics. The entropy S is 0.367 kJ/(kg•K) at 298 K under the standard pressure, while at 373 K the entropy S is 1.303 kJ/(kg•K) when the water is

liquid (the value is used for Gibbs free energies calculation in tri-phasic systems at 373 K) and 7.361 kJ/(kg•K) when the water is in the gas phase (the value is used for Gibbs free energies calculation in bi-phasic systems at 373 K).

The solvent effect was considered through the CoO surface facet modification with the H and OH groups to simulate the CoO status in the water molecule's environment. Moreover, the solvent effect was also considered implicitly through the formula $\Delta G = \Delta E + \Delta E_{ZPE} - T\Delta S + \Delta G_{pH} + \Delta G_{sol}$. At different pH values, $\Delta G_{pH} = 0.059 \times pH$; and ΔG_{sol} represents the correction term for solvent effect (0 eV for H* and 0.5 eV for OH*) (J. Mater. Chem. A, 2019, 7, 3648). As shown in Figures RRR3-RRR5, it is easily noticed that the adsorption process of water molecules in the bi-phasic system is improved, compared to that in the tri-phasic system, which favors the enhancement on the photocatalytic hydrogen evolution reaction.

In the revised version, the details on the calculation of Gibbs free energies in the bi- and tri-phasic systems and solvent effect are supplemented in the Theoretical Calculation section.

6) In order to determine some of the Gibbs free energies, the authors apply a technique now known as the computational hydrogen electrode, by J. Norskov and J. Rossmeisl (eq. 5 in the present work). While this is indeed state of the art, I find it mildly suspicious that the authors neglect to cite any of the original works here.

Response: Thanks for the comment and sorry for our carelessness. It was not intended to omit the originality. The original works from J. Norskov and J. Rossmeisl

(Chem. Phys., 2005, 319, 178; J. Phys. Chem. B 2004, 108, 17886) have been cited as Refs. 69 and 70 in the revised manuscript.

7) Finally, not a point directly related to the first-principles calculations, but with eq. 1 the authors calculate the effective free energy barrier of their reaction, yet fail to use it for anything than mildly extrapolating towards higher temperatures. Why is this barrier not discussed or at least compared to other setups? I am sure the kinetic barriers on pristine CoO can by now be found in the literature.

Response: Thanks very much for the valuable comments. Before we discuss the barrier, please allow me to discuss why we used Eq. 1. The primary purpose of Eq. 1 in the current work is to calculate the H₂ evolution rate at 373 K by fitting the experimental results between H₂ evolution reaction rate V and reaction temperature T (from 298 K to close to 373 K) in the triphase reaction system. According to Eq. 1, the H₂ evolution rate at 373 K is estimated to be 2236.76 μmol/h/g. However, it is much lower than the experimentally obtained H₂ evolution rate (6200.42 μmol/h/g, see Fig. 3d) in the biphasic reaction system at the same temperature of 373 K, indicating that in addition to reaction temperature, the state of water plays a crucial role in enhancing the hydrogen evolution of the biphasic reaction system.

Regarding the barrier, the activation energy for the hydrogen production over the CoO NPs was deduced to 23.023 kJ/mol after we used Eq. 1 to fit the experimental results between H₂ evolution reaction rate V and reaction temperature T. The activation energy of CoO NPs is lower than that of Co-CoO_x-graphene with ammonia

borane (62.3 kJ/mol) (Chem. Asian J. 2020, 15, 1) and CoO-NRs with NaBH₄ (27.4 kJ/mol) (Appl. Catal. A-Gen. 2020, 589, 117303). The activation energy is a key indicator to reflect whether photocatalytic hydrogen evolution reaction becomes easier. The smaller the activation energy is, the easier the hydrogen production process will become. Therefore, the hydrogen production process is easily conducted for the current CoO NPs.

In the revised version, the related discussion about the Eq. 1 and reaction activation energy are supplemented in the main text.

REVIEWER COMMENTS

Reviewer #1 (Remarks to the Author):

The author addressed all my technical comments in the revised manuscript to my satisfaction. With the added experimental data and analysis the authors also improved the level of scientific advancement presented in this work. The consideration of publishing this revised manuscript is thus recommended.

Reviewer #2 (Remarks to the Author):

This is the second time that I am reviewing this work. The authors significantly improved the article based on the comments of the reviewers. Still not all comments have been sufficiently addressed and as such there are still some concerns that require revisions before the article might be acceptable for publication in Nat. Commun.

- In the abstract the authors now mention impressive hydrogen production rate up to 85604 $\mu\text{mol/g/h}$. First I still suggest to be concise throughout the manuscript and use (if at all) rates normalized per area exposed as any scaling of the technology will be done using sheet like structures. Second the system providing these H₂ rates is NOT stable and as such I consider this to be inappropriate for the abstract. Given that most of the article is focused on the somewhat stable CoO/wood system I would advise to highlight this finding in the abstract
- It is noted that the hydrogen production decreases not only from day 1 to day 5 but also the accumulated amount of hydrogen throughout 8h of operation is decreasing. This should be discussed in the manuscript and it is important to emphasize that all rates are initial and not average rates obtained during 8h of illumination. Also it is still not clear what is causing the overall decrease to 90% of H₂ production already after 5 cycles. This rate of deactivation is still sufficient to consider the material unstable.
- It is still mandatory to use AQY/QY especially for the comparison performed in Fig. 5e. Given that photocatalyst dispersion is affected by anchoring to the wood substrate and in turn light scattering, particle etc is different to normal liquid phase systems using suspended particles. The variation in reactor and process design renders the comparison only meaningful if AQY/QY are used.
- The authors argue that H₂O₂ production as shown in Figure RR1 (plus evolved oxygen shown elsewhere) satisfy stoichiometric production of H₂ and "O"-species. Based on the numbers provided this is not understandable and additional details/calculations should be provided to unambiguously show stoichiometric production
- The provided stability analysis is still superficial: Particle detachment should be studied appropriate sensitive tools like elemental analysis rather than UV-VIS.

Reviewer #3 (Remarks to the Author):

The authors addressed most of my points. I am now convinced that most of their methodology is quite sound. There are only two approximations left which are still not discussed by the authors and they both concern the solvent corrections.

1) The values taken for the solvent corrections were not actually computed in reference 71. Instead reference 71 (in the supporting information, from which the two sentences in the lines 441 and 442 were copied verbatim) cites another work (Acs Catal, 2015, 5, 6658-6664) which did indeed perform calculations to estimate the free energy correction of adsorbates due to solvent. Yet, these corrections were calculated for metal particles on graphene (the original work) and then used for extended platinum in 71. Considering that the structuring of water on graphene, platinum and on oxides can be radically different (see for example any work by A. Gross and co-workers), I would think that the

direct usage of these values quite a large approximation (already in reference 71), which at least needs to be discussed.

2) In calculating the overall Gibbs free energies for the bi- and triphasic systems the authors only vary the direct entropic contribution neglecting that the solvation correction most definitely depends on the state of the solvent. This is quite easy to see if one considers that liquid water's dielectric permittivity is a lot larger than that of water vapor. The omission of this effect might explain the large differences in the free energy profiles of the two systems.

Both those points are quite strong simplifications which could have been easily avoided had the authors used e.g. an implicit solvation model in their VASP calculations. I suggest that the authors either discuss their approximations properly or avoid them altogether. As it is, I still cannot support the manuscript's publication.

Point-by-point Response to the Reviewers' Comments

Reviewer #1 (Remarks to the Author):

The author addressed all my technical comments in the revised manuscript to my satisfaction. With the added experimental data and analysis the authors also improved the level of scientific advancement presented in this work. The consideration of publishing this revised manuscript is thus recommended.

Response: We thank the reviewer for the encouragement.

Reviewer #2 (Remarks to the Author):

This is the second time that I am reviewing this work. The authors significantly improved the article based on the comments of the reviewers. Still not all comments have been sufficiently addressed and as such there are still some concerns that require revisions before the article might be acceptable for publication in Nat. Commun.

Response: We thank the reviewer for reviewing and providing insightful comments to improve the manuscript's quality.

- In the abstract the authors now mention impressive hydrogen production rate up to 85604 $\mu\text{mol/g/h}$. First I still suggest to be concise throughout the manuscript and use (if at all) rates normalized per area exposed as any scaling of the technology will be done using sheet like structures. Second the system providing these H_2 rates is NOT stable and as such I consider this to be inappropriate for the abstract. Given that most of the article is focused on the somewhat stable CoO/wood system I would advise to highlight this finding in the abstract.

Response: We thank the reviewer very much for the valuable suggestions. The hydrogen production rate unit " $\mu\text{mol h}^{-1} \text{g}^{-1}$ " has been replaced by " $\mu\text{mol h}^{-1} \text{cm}^{-2}$ " throughout the manuscript. For example, 85604 $\mu\text{mol h}^{-1} \text{g}^{-1}$ has been converted to be 3271.49 $\mu\text{mol h}^{-1} \text{cm}^{-2}$. We have also revised the abstract to specify the wood/CoO system as below: "As a result, an impressive hydrogen production rate up to 220.74 $\mu\text{mol h}^{-1} \text{cm}^{-2}$ in the particulate photocatalytic systems has been achieved based on the wood/CoO system, demonstrating that the photothermal-photocatalytic biphasic

system is cost-effective and greatly advantageous for practical applications.”

- It is noted that the hydrogen production decreases not only from day 1 to day 5 but also the accumulated amount of hydrogen throughout 8h of operation is decreasing. This should be discussed in the manuscript and it is important to emphasize that all rates are initial and not average rates obtained during 8h of illumination. Also it is still not clear what is causing the overall decrease to 90% of H₂ production already after 5 cycles. This rate of deactivation is still sufficient to consider the material unstable.

Response: We thank the reviewer for the useful comments. The description of hydrogen production has been revised as “On day 1, the initial hydrogen production rate in 1 h reaction is 221.56 $\mu\text{mol h}^{-1} \text{cm}^{-2}$, and the average hydrogen production rate during 8 h reaction is 194.14 $\mu\text{mol h}^{-1} \text{cm}^{-2}$. On day 5, the average hydrogen production rate during 8 h reaction is 174.73 $\mu\text{mol h}^{-1} \text{cm}^{-2}$. Thus, after 5 days (40 hours) test, the photocatalytic stability can be significantly improved through the wood/catalysts system compared to that in the previous work, which only holds 1 h of reaction²⁶.”

Regarding the CoO stability issue, many researchers attributed the CoO deactivation to the corroded or oxidized surfaces of CoO during the photocatalytic reaction. For example, Bao et al. found that the CoO becomes seriously deactivated after only 1 hour of reaction due to the corroded or oxidized surfaces of CoO, demonstrating that the CoO is relatively unstable (Nature Nanotechnology, 2014, 9, 69). However, in our current work, the photocatalytic stability of the CoO has been significantly improved. Even in 40 hours of reaction, the amount of H₂ production

maintains 90 % for the designed wood/CoO system. The possible reason for this is that the wood/CoO system with water steam provides a mild reaction condition relative to the triphase system with liquid water (J. Electrochem. Soc., 2020, 167, 066502; ACS Appl. Mater. Interfaces, 2019, 11, 41267)

This related discussion has been added on pages 7 and 8 of the main text in the revised version.

- It is still mandatory to use AQY/QY especially for the comparison performed in Fig. 5e. Given that photocatalyst dispersion is affected by anchoring to the wood substrate and in turn light scattering, particle etc is different to normal liquid phase systems using suspended particles. The variation in reactor and process design renders the comparison only meaningful if AQY/QY are used.

Response: Thanks very much for the valuable suggestion. The apparent quantum yield

(AQY) is calculated based on the formula below (Energy Environ. Sci. 2017, 10, 1643):

$$AQY = \frac{2 \cdot n \cdot N_A}{(E \cdot A \cdot T \cdot \lambda) / (h \cdot C)} \times 100\% \quad (1)$$

where, n is the H₂ yield, N_A is the Avogadro number, E is light intensity, A is the irradiation area, T is the time, λ is the wavelength, h is the Planks constant, C is the speed of light. The comparison of AQY with literature in different particulate photocatalytic systems has been modified, as shown in Figure R1. And the measured data is listed in Table R1. The wood/photocatalyst biphasic photothermal-photocatalytic systems show the dominant AQY in each catalyst field.

In the revised version, Figure 5e has been modified for the catalytic performance comparison based on AQY in the main text. And Table R1 is added as Supplementary Table 2 in the Supporting Information.

Figure R1. Comparison of the AQY with literature in different particulate photocatalytic systems of TiO₂, C₃N₄, MoS₂, and Co-based photocatalysts. The numbers are the wavelength information. Note: the maximum AQY data in literature and measurement results are presented for comparison.

Table R1. The measurement information for AQY calculation, including catalysts, wavelength, light intensity, and H₂ yield.

Catalysts	Wavelength (nm)/ Light intensity (mW cm ⁻²)	H ₂ yield (μmol)/ AQY (%)
Wood/CoO	380/13.82	255.01/41.1
Wood/CoO	420/16.32	405.85/50.1
Wood/CoO	500/12.32	325.77/44.8
Wood/MoS ₂	380/13.77	195.87/31.7
Wood/MoS ₂	420/16.59	295.25/35.9
Wood/MoS ₂	500/11.97	223.16/31.5
Wood/C ₃ N ₄	380/13.63	122.67/20.0
Wood/C ₃ N ₄	420/16.12	273.49/34.2
Wood/C ₃ N ₄	500/12.43	175.35/23.9
Wood/TiO ₂	360/13.54	193.74/33.6
Wood/TiO ₂	380/16.87	90.24/11.9

- The authors argue that H₂O₂ production as shown in Figure RR1 (plus evolved oxygen shown elsewhere) satisfy stoichiometric production of H₂ and “O”-species. Based on the numbers provided this is not understandable and additional details/calculations should be provided to unambiguously show stoichiometric production.

Response: We thank the reviewer for the comments. The calculation details are provided below.

Regarding the triphase system (Figure R2, i.e., Supplementary Figure 15 in the original version), after the 120-minute test, the amount of H₂ evolution is about 196.98 μmol, and the amount of O₂ evolution is approximately 48.04 μmol. After the reaction, the H₂O₂ concentration is 1.01 μmol ml⁻¹ in 100 ml reaction solvent (note that the H₂O₂ concentration is calculated through the absorption spectrum, Figure R3, i.e., Supplementary Figure 17 in the original version). Thus the amount of H₂O₂ is 101 μmol. H₂O₂ converts to O₂ following the reaction equation below:

Combining the amount of directly produced O₂ (48.04 μmol) and the amount of indirectly converted O₂ from H₂O₂ (50.50 μmol), the total amount of O₂ is 98.54 μmol. Thus, the production ratio of H₂ and O₂ is 196.68:98.54 (i.e., 1.99:1).

For the wood/CoO system (Figure R4, i.e., Supplementary Figure 16 in the original version), the amount of H₂ evolution is about 3677.55 μmol, and the amount of O₂ evolution is about 1121.21 μmol after 120 min test. The H₂O₂ concentration is

13.48 $\mu\text{mol ml}^{-1}$ in 100 ml reaction solvent after the catalytic reaction (Figure R3, i.e., Supplementary Figure 17 in the original version). Thus, the amount of H_2O_2 is 1348 μmol . Combining the amount of directly produced O_2 (1121.21 μmol) and the amount of indirectly converted O_2 from H_2O_2 (674 μmol), the total amount of O_2 is 1795.21 μmol . Considering the above calculation, the production ratio of H_2 and O_2 is 3677.55:1795.21 (i.e., 2.04:1).

In the revised version, this calculation details about stoichiometric production have been added in the Supporting Information.

Figure R2 (i.e., Supplementary Figure 15 in the original version). Time-dependent photocatalytic gas production profiles from the triphase system. The photocatalyst is CoO NPs.

Figure R3 (i.e., Supplementary Figure 17 in the original version). The H_2O_2 concentration determination from absorption spectra. (a) the absorption spectra from

different H_2O_2 concentration, and (b) the linear fitting between absorption peak and H_2O_2 concentration. The calculation details of H_2O_2 concentration: The $5\ \mu\text{mol ml}^{-1}$ H_2O_2 was used as the standard sample to react with copper (II) sulfate solution and 2,9-dimethyl-1,10-phenanthroline (DMP) solution, and the obtained absorption spectrum was used as the baseline. Due to the linear relationship between absorption peak and H_2O_2 concentration, the H_2O_2 concentration in the solution after the photocatalytic reaction could be calculated based on the above absorption spectra. After the photocatalytic reaction, the concentration of H_2O_2 measured is about 1.01 and $13.48\ \mu\text{mol ml}^{-1}$ for the triphase reaction system and wood/CoO systems, respectively.

Figure R4 (i.e., Supplementary Figure 16 in the original version). Time-dependent photocatalytic gas production profiles from the wood/CoO.

- The provided stability analysis is still superficial: Particle detachment should be studied appropriate sensitive tools like elemental analysis rather than UV-VIS.

Response: We thank the reviewer for the valuable suggestion. The photocatalytic particle stability has been further studied through X-ray photoelectron spectroscopy (XPS) and inductively coupled plasma emission spectrometer (ICP). As shown in Figure R5, the XPS spectra of the wood/CoO system maintain relatively unchanged after the catalytic reaction. The high-resolution element spectra of elemental C, Co,

and O have also been displayed in Figure R5, exhibiting little change before and after the catalytic reaction. Besides, the ICP of the bulk water in the wood/CoO system after the photocatalytic reaction has also been measured, and a trace amount of element Co in the bulk water is observed, indicating the excellent particle stability of the wood/CoO system (Table R2).

In the revised version, the XPS and ICP results have been added as Supplementary Figure 13 and Supplementary Table 1 in the Supporting Information. The related discussion has been added on page 8 of the main text.

Figure R5. The XPS spectra of the wood/CoO system before and after photocatalysis process. **a** XPS full spectrum of the wood/CoO system before/after the reaction, **b** high-resolution XPS of element O, **c** element C, and **d** element Co for the wood/CoO system.

Table R2. The element Co concentration in the bulk water from the wood/CoO

system after the photocatalytic reaction based on the ICP measurement. Three samples were tested.

	Sample 1	Sample 2	Sample 3
Element Co Concentration ($\mu\text{g ml}^{-1}$)	0.0102	0.0123	0.0088

Reviewer #3 (Remarks to the Author):

The authors addressed most of my points. I am now convinced that most of their methodology is quite sound. There are only two approximations left which are still not discussed by the authors and they both concern the solvent corrections.

1) The values taken for the solvent corrections were not actually computed in reference 71. Instead reference 71 (in the supporting information, from which the two sentences in the lines 441 and 442 were copied verbatim) cites another work (Acs Catal, 2015, 5, 6658-6664) which did indeed perform calculations to estimate the free energy correction of adsorbates due to solvent. Yet, these corrections were calculated for metal particles on graphene (the original work) and then used for extended platinum in 71. Considering that the structuring of water on graphene, platinum and on oxides can be radically different (see for example any work by A. Gross and co-workers), I would think that the direct usage of these values quite a large approximation (already in reference 71), which at least needs to be discussed.

2) In calculating the overall Gibbs free energies for the bi- and triphasic systems the authors only vary the direct entropic contribution neglecting that the solvation

correction most definitely depends on the state of the solvent. This is quite easy to see if one considers that liquid water's dielectric permittivity is a lot larger than that of water vapor. The omission of this effect might explain the large differences in the free energy profiles of the two systems.

Both those points are quite strong simplifications which could have been easily avoided had the authors used e.g. an implicit solvation model in their VASP calculations. I suggest that the authors either discuss their approximations properly or avoid them altogether. As it is, I still cannot support the manuscript's publication.

Response: We thank the reviewer very much for the valuable suggestion. The two questions are both related to solvent corrections and can be answered together. We agreed with the reviewer and have chosen the implicit solvation model to avoid the simplification problem of solvent corrections because the implicit solvation model considers the effect of water's dielectric permittivity. The implicit solvent models are calculated using the VASPsol (J. Chem. Phys. 2014, 140, 084106; J. Chem. Phys. 2019, 151, 234101), where the dielectric constants of liquid water and gas water are indexed to be 81 and 1, respectively from the Handbook of Chemistry and Physics. As a result, the Gibbs energy of a photocatalytic reaction can be calculated to better explain the bi- and triphasic systems' differences, as shown in Figure R6. Compared to that in the tri-phasic system, it is easily noticed that the water molecule adsorption process in the bi-phasic system is much more favorable to the photocatalytic hydrogen evolution reaction. It is concluded that the calculation results with the implicit solvation model still support our findings.

The details on the calculation of implicit solvent models are supplemented in the Theoretical Calculation section in the revised version. And the related discussion is added in the main text.

Figure R6. **a** Gibbs energy of a photocatalytic reaction on the pure CoO surface at different temperatures in the triphase system. **b** Gibbs energy comparison of a photocatalytic reaction on the pure CoO surface at 373 K in bi- and triphasic systems.

REVIEWERS' COMMENTS

Reviewer #2 (Remarks to the Author):

The author addressed all my comments in the revised manuscript. The revised manuscript is thus recommended for publication.

Reviewer #3 (Remarks to the Author):

The authors addressed my last point. I am now convinced that their theoretical results are based on a sound state of the art methodological approach. I have found no further reasons preventing the publication of this manuscript in Nature Communications.

Point-by-point response to reviewers' comments

Reviewer #2 (Remarks to the Author):

The author addressed all my comments in the revised manuscript. The revised manuscript is thus recommended for publication.

Response: Thanks very much for the comments.

Reviewer #3 (Remarks to the Author):

The authors addressed my last point. I am now convinced that their theoretical results are based on a sound state of the art methodological approach. I have found no further reasons preventing the publication of this manuscript in Nature Communications.

Response: Thanks very much for the comments.